# The cellular environment shapes the nuclear pore complex architecture

Anthony P. Schuller[1,7], Matthias Wojtynek[2,3,7], David Mankus[4], Meltem Tatli[2], Rafael Kronenberg-Tenga[2], Saroj G. Regmi[5], Phat V. Dip[1,6], Abigail K. R. Lytton-Jean[4], Edward J. Brignole[1,6], Mary Dasso[5], Karsten Weis[3], Ohad Medalia[2✉] & Thomas U. Schwartz[1✉]

Nuclear pore complexes (NPCs) create large conduits for cargo transport between the nucleus and cytoplasm across the nuclear envelope (NE)[1–3]. These multi-megadalton structures are composed of about thirty different nucleoporins that are distributed in three main substructures (the inner, cytoplasmic and nucleoplasmic rings) around the central transport channel[4–6]. Here we use cryo-electron tomography on DLD-1 cells that were prepared using cryo-focused-ion-beam milling to generate a structural model for the human NPC in its native environment. We show that—compared with previous human NPC models obtained from purified NEs—the inner ring in our model is substantially wider; the volume of the central channel is increased by 75% and the nucleoplasmic and cytoplasmic rings are reorganized. Moreover, the NPC membrane exhibits asymmetry around the inner-ring complex. Using targeted degradation of Nup96, a scaffold nucleoporin of the cytoplasmic and nucleoplasmic rings, we observe the interdependence of each ring in modulating the central channel and maintaining membrane asymmetry. Our findings highlight the inherent flexibility of the NPC and suggest that the cellular environment has a considerable influence on NPC dimensions and architecture.

A combination of X-ray crystallography and electron microscopy (EM) approaches has generated a consensus model for the NPC architecture, characterized by three substructures that span the NE[4,5,7]. The inner ring (IR) is centrally positioned at the fusion point of the inner and outer nuclear membranes (INM and ONM, respectively) and forms the central channel that connects to phenylalanine–glycine (FG) repeat proteins that create the permeability barrier and facilitate active cargo transport. The cytoplasmic ring (CR) provides a platform for protein and RNA export[3,8]. The nucleoplasmic ring (NR) shares scaffold nucleoporins (Nups) with the CR and anchors a distinct basket-like structure with diverse functions.

Although all eukaryotes maintain a three-ring NPC structure, the composition and arrangement of the substructures varies between species. In mammalian cells, the IR has five primary subunits: Nup205, Nup188, Nup155, Nup93 and Nup35 (refs. [4,5,9]). FG-Nups (Nup54, Nup58 and Nup62) are docked to the IR to create the central channel for transport[4,5,7]. The CR and NR both contain 16 copies of the Y complex[10–13], which contains a set of elongated helical stack proteins (Nup160, Nup133, Nup107, Nup96 and Nup85) decorated with the beta-propellers Sec13, Seh1, Nup37 and Nup43 (refs. [11,12,14]). The principal Y-complex arrangement in both the CR and NR follows a reticulated pattern composed of two eight-membered rings[12,15].

Our current understanding of the human NPC structure is based mainly on cryo-electron tomography (cryo-ET) studies of NPCs from purified NEs at 2–6 nm resolution[12,15,16]. Cryo-focused-ion-beam

(cryo-FIB) milling now enables the study of macromolecular complexes in their native environment[17]. NPCs from cryo-FIB-milled yeast[18,19], algae[20] and human cells[21,22] show differences compared with previous human NPC structures, including wider IRs. Here we study the architecture of the human NPC from cryo-FIB-milled DLD-1 cells. Our data establish concepts in human NPC architecture that may have previously been interpreted as species-specific differences. Our study shows that the cellular environment substantially influences the diameter of the NPC central channel and emphasizes the modular, although interdependent, architecture of the NPC and its role in shaping the NE.

## The architecture of the NPC in DLD-1 cells

We used cryo-FIB-milled lamellae of human DLD-1 cells containing an auxin-inducible degron (AID) tag at the *NUP96* (HGNC symbol: *NUP98*) loci (homozygous *NUP96::Neon-AID*)[23] for the targeted depletion of Nup96. Such Nup96 depletion leads to a loss of both the CR and NR elements, whereas the IR is unaffected. Here we compared native and partially degraded NPC structures. Lamellae were prepared from cells in the absence of auxin (wild-type condition) for cryo-ET imaging and identification of in situ NPCs. We extracted 194 NPC-containing subvolumes from 54 tomograms and used subtomogram averaging to obtain a density map of the NPC resolved to ~3.4 nm (Fig. 1 and Extended Data Fig. 1a–c). Our structural analysis revealed a similar three-ring architecture to that observed in previous human models[12,15,16]. However, the diameter of

[1]Department of Biology, Massachusetts Institute of Technology, Cambridge, MA, USA. [2]Department of Biochemistry, University of Zurich, Zurich, Switzerland. [3]Department of Biology, Institute of Biochemistry, ETH Zurich, Zurich, Switzerland. [4]Koch Institute for Integrative Cancer Research, Massachusetts Institute of Technology, Cambridge, MA, USA. [5]Division of Molecular and Cellular Biology, National Institute of Child Health and Human Development, NIH, Bethesda, MD, USA. [6]MIT.nano, Massachusetts Institute of Technology, Cambridge, MA, USA. [7]These authors contributed equally: Anthony P. Schuller, Matthias Wojtynek. ✉e-mail: omedalia@bioc.uzh.ch; tus@mit.edu

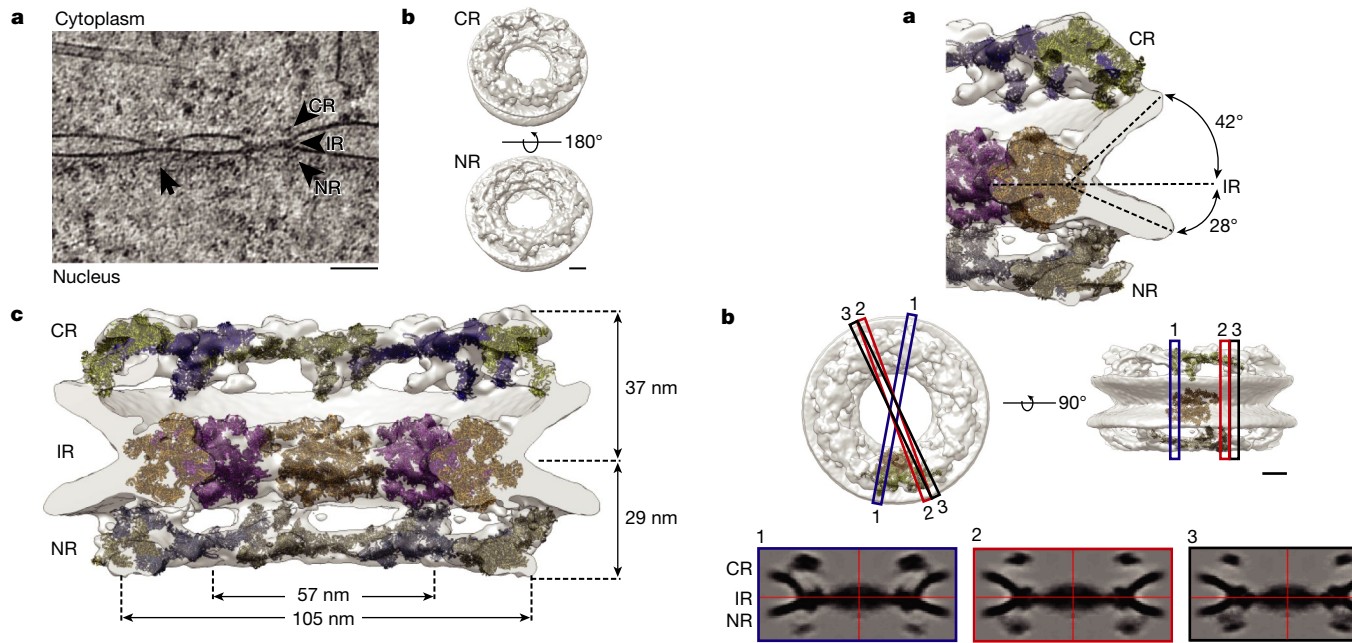

**Fig. 1 | The NPC structure from human DLD-1 cells. a**, Tomographic slice from a lamella of a *NUP96::Neon-AID* DLD-1 cell in the absence of auxin (wild-type condition). The slice thickness is 2.7 nm. The arrowheads indicate each of the NPC rings. The arrow indicates an adjacent NPC. Scale bar, 100 nm. **b**, Whole NPC map from DLD-1 cells. Scale bar, 20 nm. **c**, Cross-section of a cryo-ET map with fitted structures for the CR (blue and yellow), IR (orange and magenta) and NR (light blue and gold). The central channel diameter (57 nm), the membrane-to-membrane diameter (105 nm) and the height across the NE (37 nm from IR-CR, 29 nm from IR-NR) are indicated.

**Fig. 2 | Asymmetric nuclear membrane at the NPC. a**, Cross-section of the NPC structure. The angles indicate measurements from the membrane distal edge of the IR to the edge of the nuclear membrane at the NR or CR. **b**, 2D orthoslices through the full-NPC average. Top, NPC map; the planes of the 2D slices (bottom) are indicated. Scale bar, 20 nm. Plane 1, at the point of the interaction between the CR Y complex and the membrane. Plane 2, at the point of the interaction between the NR Y complex and the membrane. Plane 3: sSlice in which no Y complexes touch membrane. The slice thickness is 1.4 nm.

the central channel is considerably different (Fig. 1c and Extended Data Fig. 2a)—it is about 33% larger (width, about 57 nm) compared with previous models (width, about 43 nm)[15]. Furthermore, the outer diameter of the IR is about 105 nm, compared with around 89 nm for semi-purified NPCs. Moreover, we observed a reduced NPC height of 66 nm, compared with 75 nm in previous models. Finally, the distance between the centre of the IR and the distal end of the CR is greater than the distance between the IR and the NR (37 nm versus 29 nm, respectively) (Fig. 1c).

We fit structures of nucleoporin subcomplexes into the three rings of the averaged map (Fig. 1c and Extended Data Fig. 1d–f). For the NR and CR, we used a composite Y-complex model derived from overlapping crystal structures and threaded models[11,14,15,24]. Bending at recognized 'hinge points'[10,11,14], we fit the Y complex (Extended Data Fig. 3a–c). It docks into both the CR and NR in a reticulated pattern of eight-membered rings (Extended Data Fig. 4a–e). In each ring, we observed considerable density beyond the Y complexes. The additional density in the CR might correspond to Nup358, an expected component of this subassembly[15] (Extended Data Fig. 4b). We also observed density protruding out of the CR towards the centre of the ring at the base of Nup85 (Extended Data Fig. 1d). On the basis of previous cross-linking[8,25] and cryo-EM/cryo-ET[26,27] studies, this density is probably occupied by the Nup214–Nup88–Nup62 complex.

For the IR, we first docked the published composite model containing Nup93, Nup188/Nup205, Nup155 and the Nup62–Nup58–Nup54 complex[9] (Extended Data Fig. 3d, e). Although this model fits well, some elements appear to be shifted, suggesting that flexibility in the IR Nups may account for the larger diameter that we observed (Extended Data Fig. 5a). To better fit the model, we isolated individual Nups and reoriented them locally to create an updated IR model (Extended Data Fig. 5a, b).

The NR is also composed of two eight-membered rings of Y complexes. As in the CR, we identified additional density near the Nup85 arms (Extended Data Fig. 1f), and observed a linker density between

the Y rings in agreement with a previous report[15] (Extended Data Fig. 4c). Unique to the NR, we observed density at the base of the Y-complex-containing rings, which extends towards the IR (Extended Data Fig. 5e, f).

## Asymmetric nuclear membranes at the NPC

Cross-sections through the cryo-ET map reveal that the NE at the NPC is asymmetric, with a steeper angle at the CR compared with at the NR (about 42° compared with about 28°, respectively), using the IR as a plane of reference (Fig. 2a). This membrane asymmetry is also evident in two-dimensional (2D) orthoslices of eight-fold symmetrized NPCs. In different planes of the NPC, it is readily apparent how the NE is shallower at the nucleoplasmic versus the cytoplasmic surface (Fig. 2b).

## Native NPC has a larger central channel

Compared with previous human NPC models, our map displays substantial differences, including larger IR and central channel diameters. Although the IR is essential for maintaining the architecture of the NPC central channel, it also anchors Nups with flexible, unstructured FG-rich domains that facilitate transport through the channel[7,28]. We found that the channel delineated by the IR has a volume that is about 75% larger than previously reported[15] (Fig. 3a). We expect that these differences are due to the preserved cellular environment rather than any cell-type-specific differences, as we observed a similar architecture in HeLa-derived cells (Extended Data Fig. 2a, b). Furthermore, we analysed data from mouse embryonic fibroblasts in which cryo-ETs of NPCs were captured by cryo-FIB milling or cell permeabilization followed by nuclease incubation[29]. The diameter of NPCs from cryo-FIB cells was again notably larger compared with the diameter of NPCs from permeabilized cells (Extended Data Fig. 2c).

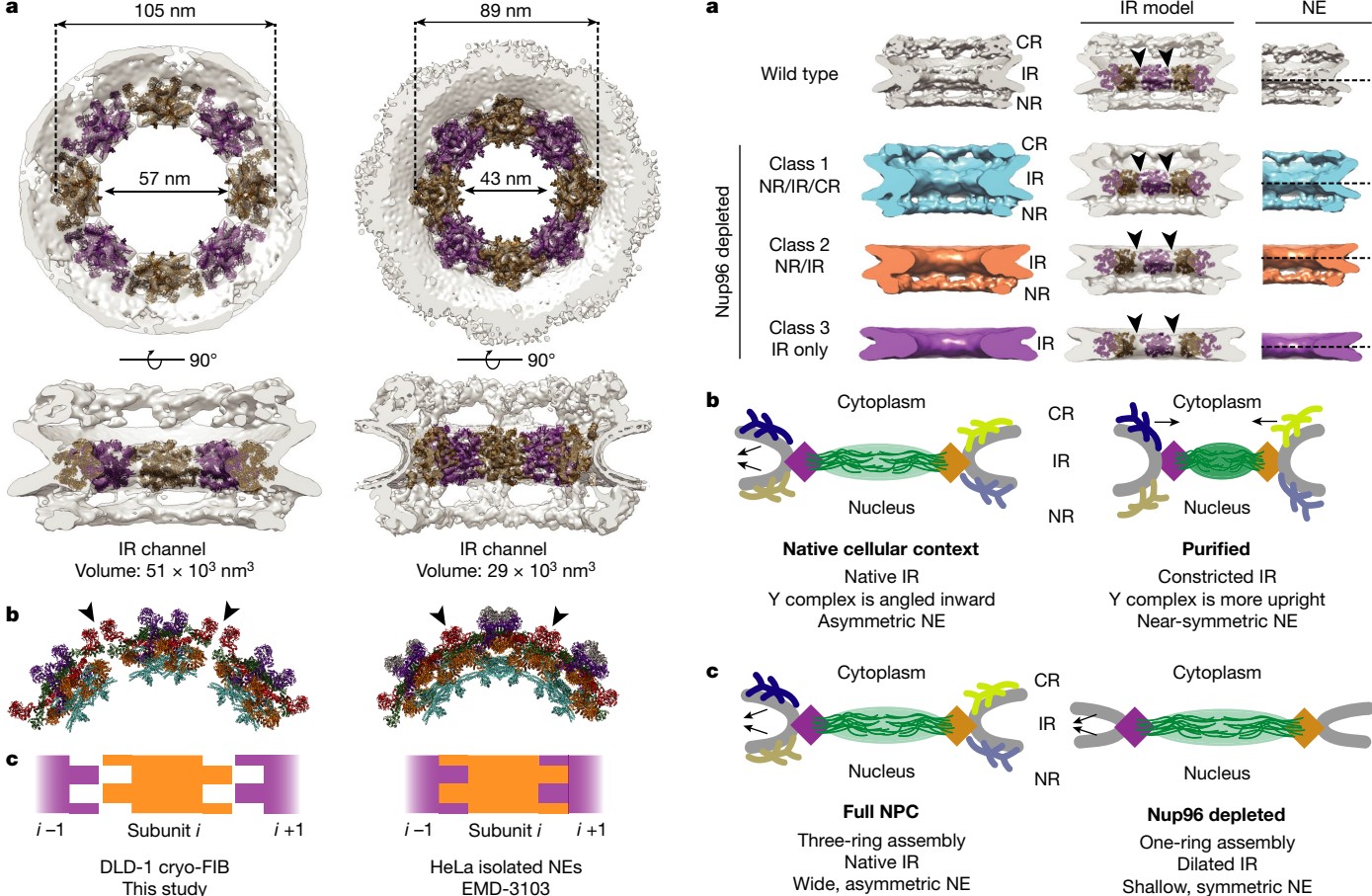

**Fig. 3 | IR flexibility increases central channel size in native NPCs.**
**a**, Segmentation of the IR from our model and the previous human NPC model. The IR diameter (membrane to membrane) and central channel diameter measurements are indicated. The IR channel volume was calculated as a cylinder with a height of 20 nm. **b**, Top view of three IR subunits in our model compared with the previous model. **c**, Schematic showing the lock-and-key-like architecture of the previous model and how the IR subunits have shifted in our model.

In our IR model, Nup205/Nup188 (paralogues that are indistinguishable at this resolution), Nup155 and Nup93 are significantly shifted, whereas the Nup62–Nup58–Nup54 complex is stable (Extended Data Fig. 5b). Propagated through the entire IR, these rearrangements explain how the central channel becomes larger. Adjacent IR subunits are nearly separated compared with previous models in which the subunits were snugly aligned (Fig. 3a, b and Extended Data Fig. 5c, d). The Nup93 and Nup205 subunits of adjacent IR structures form a pseudo-lock-and-key configuration in the constricted IR model; by contrast, here these components shifted to create gaps between the IR subunits (Fig. 3c).

In contrast to the IR, the diameters of the CR and NR are nearly identical to previous models (Extended Data Fig. 4d, e). However, to accommodate a flatter overall assembly, the CR and NR rings flex in relation to the NE. At both the CR and NR, the Y-complex rings are angled more inward towards the central channel (Extended Data Fig. 4f).

## Architecture of Nup96-depleted NPCs

To better understand the contribution of each ring to the overall NPC assembly, we next analysed the structure of Nup96-depleted NPCs. We acquired 71 tomograms of cryo-FIB-milled *NUP96::Neon-AID* DLD-1 cells after auxin-induced Nup96 depletion and extracted 163 NPC-containing subvolumes for subtomogram averaging (Extended Data

**Fig. 4 | Ensemble of Nup96-depleted structures reveal the interdependence of the NPC rings and role of the rings in shaping the nuclear membrane.**
**a**, NPC structures from *NUP96::Neon-AID* DLD-1 cells in the presence of auxin (Nup96-depleted) compared with wild-type (grey). Class 1 (blue, *n* = 27) contains all three rings. Class 2 (orange, *n* = 53) contains the NR and IR substructures. Class 3 (purple, *n* = 83) contains only the IR. A comparison of IR model docking (middle), and NE curvature (right) is included. The arrowheads indicate the gaps between adjacent IR subunits. The dashed line is drawn at the midplane of the IR in the cross-sections. **b**, Model of the structural differences between our native NPC structure and the previous structure from purified NEs. **c**, Model of the structural changes caused by Nup96 depletion.

Fig. 6a–f). In two cells, we observed ONM herniations (Extended Data Fig. 6g), indicating dysfunctional assembly of NPCs[30]. NPCs from this dataset could be divided into the following three classes (Fig. 4a): class 1, with all three rings (17% of particles); class 2, with NR–IR complexes (32%); and class 3, containing only the IR (51%). As we found no CR–IR complexes, this suggests that the CR is more readily removed from the NPC than the NR after Nup96 depletion. Furthermore, the diameters of each unique NPC assembly varied, and assemblies with NR–IR or IR only had significantly larger diameters (Extended Data Fig. 6c). With the reduction from three to two to one ring, the spacing between the eight IR subcomplexes increases (Fig. 4a). In addition to the dilated central channel in IR-only NPCs, a cross-section analysis revealed changes in membrane orientations at the opening surrounding Nup96-depleted NPCs (Fig. 4a). The membrane has a smaller radius of curvature at the ONM–INM fusion, and the NE is more symmetric compared with in wild-type NPCs.

## Discussion

The architectural analysis of the NPC is a major challenge for structural biology. To date, the most accurate models of the human

NPC are composite structures combining crystal or EM structures and structure-based models of subcomplexes with cryo-ET data of semi-purified, intact NPCs[9,15,31]. Our study reveals substantial NPC plasticity and shows that the purification of NEs influences the human NPC structure. We considered how the difference in IR diameter between in situ versus partially purified NPCs can be explained. As the NPC is integrated into the cellular matrix through the lamina and the cytoskeleton[32,33], it is conceivable that tensile forces acting on the NE could cause this change (Fig. 4b). Another consideration is that nuclear transport facilitated by FG-repeat extensions interacting with transport receptors may affect the NPC architecture, as reported in *Xenopus* oocytes[34]. Furthermore, some FG-Nups stabilize the NPC scaffold[35]; there could therefore be a yet-to-be discovered connection.

Modulating the IR diameter may be a general property of NPCs as we also observed this phenomenon in mouse embryonic fibroblasts, and it has been reported in *Saccharomyces cerevisiae*[18], *Schizosaccharomyces pombe*[19] and human cells[21,22]. Moreover, we observed heterogeneity in the IR diameter among individual NPCs. Thus, structural variation is an intrinsic property of the NPC inside cells. IR flexibility could also be important for protein trafficking to the INM, which requires passage through the NPC while membrane anchorage is retained[18,20,36–38]. A wider central channel not only has implications for the maximal size of soluble and membrane-bound cargo that can pass through the NPC, but it also changes the density of the FG repeats that fill the transport channel[39]. The influence of FG density on transport has been documented[40]. Our NPC model indicates that the FG density in the central channel is lower than previously thought, or that flexibility may be a means to modulate the FG barrier, consistent with suggestions for NPCs from *Chlamydomonas reinhardtii*[20] and *S. pombe*[19].

In contrast to the IR diameter change, the CR and NR dimensions are similar to previous studies of semi-purified NPCs, suggesting that the three rings have some independence. Our findings in Nup96-depleted cells support a modular design, but also highlight the interdependence of these modules in regulating the central channel (Fig. 4c). This may indicate that the CR and NR act as molecular 'rulers' to define the upper limit of the IR dimensions, as proposed for *C. reinhardtii*[20]. However, the fact that the IR is malleable whereas the CR and NR are not also implies that the connections between the distinct rings (IR–NR and IR–CR, respectively) are flexible, explaining different assignments in various reconstructions[9,12,15,16,18,26,36].

Our model also exposes the asymmetry of the CR and NR substructures and their effect on the NE, as previously observed in the *Xenopus laevis* NPC[26]. Furthermore, we observed asymmetry in the nuclear membrane at the NPC, suggesting that the membrane connections between the NR and CR are different. It is possible that ring-specific Nups make interactions with the cytoskeletal/laminar network or with ONM/INM-specific proteins to bring about this asymmetry. Alternatively, the ring-linking densities (such as Nup358 at the CR or the unassigned NR linker) could contort the CR/NR reticulated ring structures differentially, as their diameters are slightly different. Thus, the tension that each ring might impose on the NE may also lead to this asymmetry.

Flexibility of the NPC emerges as a major challenge for achieving true high resolution. In addition to flexibility, architectural heterogeneity might come from NPCs that are assembled from different subsets or stoichiometries of Nups[41,42]. Thus, the structure described here probably represents just one of several NPC states that are still to be discovered.

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

## Methods

### Cell culture and Nup96 depletion

*NUP96*::*Neon-AID* DLD-1 cells[23] were grown in standard tissue culture conditions (37 °C with 5% CO$_2$) using DMEM medium supplemented with 4.5 g l$^{-1}$ glucose and 4.5 g l$^{-1}$ sodium pyruvate (Corning), 10% FBS (Gemini Bio) and 2 mM GlutaMAX (Gibco). Next, cells were cultured for 18–24 h on freshly glow-discharged carbon-coated EM gold grids (200 mesh, R1.2/1.3 or R2/1; Quantifoil Micro Tools) before plunge-freezing. For Nup96 depletion, a 250 mM auxin (3-indoleacetic acid, Sigma-Aldrich) solution was added to the cells to a final concentration of 1 mM, followed by incubation at 37 °C for 4–8 h before plunge-freezing. HeLa-derived (*TOR1A/TOR1B/TOR3A*-knockout) cells[43] were cultured in the same conditions but never subjected to auxin treatment.

### Cryo-FIB milling of lamellae from DLD-1 cells

To prepare lamellae, we used both an Aquilos FIB-SEM system (Thermo Fisher Scientific) and a Crossbeam 540 (Zeiss) with cryo stage (Leica) using similar methods as previously described[44]. On the Aquilos system, grids were loaded into the sample chamber and sputtered with an initial platinum coat (15 s) followed by a second organometallic platinum protective layer using the gas injection system (GIS, 15 s). Samples were tilted to an angle of 15°, and FIB ablation of cellular material was performed in a stepwise manner by focusing the gallium beam at 30 kV on parallel rectangular patterns with over- and under-tilting to the desired thickness as follows: (1) 15°, 500 pA, gap = 3 μm; (2) 16°, 300 pA, gap = 2.2 μm; (3) 14°, 300 pA, gap = 1.5 μm; (4) 15.5°, 100 pA, gap = 0.8 μm; (5) 14.5°, 100 pA, gap = 0.5 μm; (6) 15°, 50 pA, gap = 300 nm; and (7) 15°, 30 pA, gap = 150 nm. The initial width of the lamellae was set to 13 μm but, at each reduced current, we reduced the width by 0.25 μm, for lamellae with a final width of 11–12 μm. Expansion/relief joints to reduce tension at the lamella were milled 3 μm away from the sides of the lamella at step 1 with a 1-μm-wide pattern. A similar procedure was used on the Crossbeam 540 FIB-SEM. Grids were first electron-beam-coated with 2 nm of platinum in an ACE 900 system (Leica) before being loaded into the Crossbeam sample chamber with a VCT 500 (Leica) vacuum transfer system for organometallic platinum coating. FIB ablation was performed using only four stepwise currents and without over- and under-tilting as follows: (1) 700 pA, gap = 3 μm; (2) 300 pA, gap = 1 μm; (3) 100 pA, gap = 400 nm; and (4) 50 pA, gap = 150 nm. In total, we prepared 37 lamellae from non-auxin-treated *NUP96*::*Neon-AID* DLD-1 cells, and 115 lamellae from auxin-treated cells to obtain our data.

### Tilt series acquisition and processing

Cryo-ET datasets were acquired on a Titan Krios G3i operating at 300 keV equipped with a BioQuantum post-column energy filter and a Gatan K3 direct electron detector. Tilt series were acquired at a magnification of ×26,000, resulting in a pixel size of 3.4 Å at the specimen level using Tomo v.5.3.0 (Thermo Fisher Scientific). We used a tilt range of −52° to +68°, starting at +8° with a bidirectional scheme[45] resulting in a total dose of about 145 electrons per angstrom squared. Tilt series of HeLa cells (*TOR1A/TOR1B/TOR3A*-knockout)[43] were acquired with an accumulative dose of about 120 electrons per angstrom squared. Defocus values varied between −5.0 μm to −2.5 μm through the entire data acquisition process. Representative tomograms are included (Supplementary Videos 1 and 2).

Tilt series were aligned with 4× binned projections using patch tracking in the IMOD software package[46]. When available, contaminations generated by the FIB process were used as fiducial markers. The contrast transfer function (CTF) was determined and corrected as described previously[26], and is summarized here. The mean defocus was estimated by strip-based periodogram averaging for each tilt series. Using the mean defocus, the tilt angle and axis orientation, the defocus gradient for each projection was determined. CTF correction using the defocus gradient was then performed by phase-flipping each projection image. CTF-corrected stacks were next dose-filtered using the IMOD mtffilter function[46]. NPC coordinates were picked manually using IMOD. During NPC particle picking, the relative orientation of the NPC to the NE was determined to prealign the NPCs. Next, subtomograms of NPCs were reconstructed using IMOD. Detailed imaging parameters are summarized in Extended Data Table 1.

### Subtomogram averaging

For the wild-type DLD-1 structure, 194 prealigned NPCs were further aligned using iterative missing-wedge-weighted subtomogram alignment and averaging with the TOM toolbox (tom_corr3d)[47]. Half-set averages were merged after each iteration and used as the template for the next iteration. First, full NPCs were aligned using 8× binned subvolumes and applying eightfold-fold symmetry. This step was repeated using 4× binned subvolumes. Subsequently, these alignments were used to extract the eight asymmetric units (1,552 protomers) for each NPC according to the eight-fold symmetry as previously described[48]. Twice-binned full protomers were aligned and further refined by applying masks for the individual rings (CR, IR and NR). Next, subprotomers were extracted based on the previous alignment step. At this stage, single protomers were manually inspected and missing protomers from incomplete NPCs, misaligned protomers and protomers with a low signal-to-noise ratio were excluded. After this step, 1,252 protomers remained for the subsequent steps. Newly extracted subprotomers were further aligned. Resolution was measured using the 0.5 criterion and soft masks to exclude an artificial contribution to the measured resolution using the Electron Microscopy Data Bank (EMDB) validation server (https://www.ebi.ac.uk/pdbe/emdb/validation/fsc/). Subprotomer volumes were *B*-factor sharpened using Relion[49] with a *B*-factor of −2,000 Å$^2$. The final full NPC model was generated by fitting the subprotomer volumes into the full NPC average using UCSF Chimera[50].

Averaging of auxin-treated (Nup96-depleted) data was performed similarly. First, 163 full NPCs were aligned using 8× binned subprotomers and applying eight-fold symmetry. Next, NPCs were classified by manually inspecting the subvolumes into three classes containing CR, IR and NR (27 NPCs), IR and NR (53 NPCs), or IR only (83 NPCs). These three classes were further aligned using 8× binned subvolumes at full NPC level using eight-fold symmetry.

For the HeLa (*TOR1A/TOR1B/TOR3A*-knockout) average, 27 NPCs were aligned similar to the wild-type dataset but without dose-filtering of the stacks. After protomer extraction, 36 protomers were excluded and protomers were aligned. The final map (180 protomers) was generated by fitting the protomer average back into the full NPC map using Chimera.

### Model fitting

To create models for the Y complexes in the CR and NR, we performed unbiased global fitting using structural models derived from previously reported human NPC structures[9,15]. For these complexes, we began with the previously reported model and performed a global fitting analysis as implemented in UCSF Chimera[50]. The fitting was performed independently for the CR and NR using a three-protomer segmented model as our single protomers do not necessarily contain a complete protomer of the eight-fold symmetric NPC (that is, a single protomer may contain half of the Y complex from adjacent protomers). For the IR, a single protomer was used as it contained the entire complex. All fitting runs were performed using Chimera and 1,000,000 random initial placements and local cross-correlation (Chimera's correlation about the mean (CAM)), or a combination of local cross-correlation and overlap (CAM + OVR) was used when local cross-correlation did not provide statistically significant fits (as others have observed)[20]. We computed fit scores using both metrics for each model but, for the IR, the local cross-correlation (CAM) provided statistical significance to our model, whereas, for the Y complex, the combination metric (CAM + OVR) provided statistical significance. We noted the scores

and adjusted *P* values (described below) on the figure for each scoring metric. For each fitting run, the statistical significance was assessed as a *P* value that was calculated from normalized fitting scores. To calculate the *P* values, we transformed the CAM or combined CAM + OVR scores into *Z* scores, derived a two-sided *P* value for each fit and then corrected the *P* values for multiple testing using the conservative Benjamini–Hochberg procedure. This workflow was performed using a Python script running SciPy.Stats (for *P* value and *Z*-score analysis)[51], the StatsModels module (for Benjamini–Hochberg analysis)[52] and Matplotlib (for plots)[53].

For the IR complex, we identified a significant fit in the expected position as described in the most recent human structure (Protein Data Bank (PDB): 5IJN)[9], although it was obvious that some of the Nups in this model could be shifted to better fit our density map. We therefore followed a previously described approach[20] and optimized the fits by local refitting of individual subunits or domains. For this refinement, we used the option in Chimera to use a map simulated from atoms at a resolution of 25 Å and optimized the fitting for correlation. Using this approach, we fit subcomplexes made of the chains F, G, H, X, Y and Z (two copies of Nup54, Nup58 and Nup62), L, M, N, R, S and T (two additional copies of Nup54, Nup58 and Nup62), each copy of Nup205 (chains D, J, P and V), and the four copies of Nup93 (chains C, I, O and U). To fit the Nup155 chains, we subtracted densities with a radius of 10 Å of the fitted models above from the determined NPC map. In this difference map, we fit the individual Nup155 copies into the map using the procedure described above (chains E, Q, K, W). The Nup155 chains A and B of 5IJN could not be fit into our map reliably. We next ran a new unbiased global search using this model as the input for completeness.

For the Y complex, we identified a significant fit in the same position as described in the most recent human NPC structure CR and NR (PDB: 5A9Q)[15] using a double Y complex as the reference molecule. As the 5A9Q model contains gaps in the heterotetrameric core element of the Y complex called the 'hub' (Nup160, Nup85, Nup96 and Sec13), we decided to replace the hub of the 5A9Q model with the hub of a new composite model that we created using published crystal structures and threading models. We first generated an *S. cerevisiae* composite model combining a previous structure of the Y complex hub (Nup120–Nup145N–Sec13–Nup85)[14] (PDB: 4XMM) and the Nup84–Nup133 subcomplex[24] (PDB: 6X02). This *S. cerevisiae* Y-complex composite was then used to template the human homologue, using published structures and threading models. The hub of this updated model was then used to rigid-body dock into the CR/NR of our map at the same position that the 5A9Q hub occupied, and the Nup107–Nup133 subcomplex from 5A9Q was kept as is, creating a new complete Y-complex assembly. Fitting was further refined by cutting Nup160 between residue 933 and 934 (ref. [11]) and fitting the N-terminal part of Nup160 (Nup160-N) together with Nup37 as a rigid body into our map. The C-terminal part of Nup160 was fit as part of the hub. The CR model was then also placed for the inner Y ring and not further modified. Fitting for the NR Y complex was performed similarly. For the outer Y ring, only the hub and the Nup107–Nup133 subcomplex were separated and fit as individual rigid bodies. For the IR of the NR Y complex, the model without Nup160-N and Nup37 was fit into the map. For fitting of the Nup160-N–Nup37 subcomplex, we subtracted the densities of previously fit models with a radius of 20 Å from the map and fit the subcomplex into the remaining density. After creating the final models for Y complexes in both the CR and NR, we performed another round of unbiased global fitting and identified a similar location at which the Y complex of 5A9Q docked. A final local optimization run maximized density overlap. We strongly urge that the final fits are not interpreted at atomic resolution. Instead, our fitting simply aids in assignment of our density toward understanding how each of the subcomplexes is positioned.

The IR model described above was placed manually into the centre of the IR protomers of the subtomogram maps generated from auxin-treated (Nup96-depleted) cells.

## Visualization

Visualizations were performed using UCSF Chimera[50]. Representative tomograms of DLD-1 cells were reconstructed in 8× binned, 12× SIRT-like filtered tomograms in IMOD[46]. Snapshots of mouse embryonic fibroblast (MEF) NPCs were recorded after manually aligning the NPCs in IMOD Slicer. The orthoslice views of the full-NPC average, wild-type map and auxin maps were taken using tom_volxyz. Scripts were implemented in MATLAB (MathWorks) and using the tom_toolbox[54].

Local resolution analysis was performed as previously reported[26], but also summarized here. To calculate the local resolution of the sub-protomers, the full subprotomer volume (100 × 100 × 100 voxels) was divided into smaller subvolumes (box size, 40 × 40 × 40) along a regular spacing of 4 × 4 × 4. Resolution was measured between the subvolumes of the two half sets and using the 0.5 threshold criterion. Subvolumes were masked with a spherical mask prior to FSC calculation. Data points between measured values were interpolated and visualized using Chimera's Surface color function.

## NPC diameter analysis

For diameter measurements, single NPCs were measured using ortho-slices at the level of the pore membrane. For the DLD1 and HeLa data-sets, aligned NPCs that would proceed to full-NPC averaging in this study were used. For the MEF datasets[29], NPCs were manually aligned in IMOD slicer and measured. Distance was measured between the NE at the narrowest point of each NPC manually. When possible, measurement was performed in two orthogonal directions and the average was calculated. Otherwise, only a single measurement was performed per NPC. When measurement was not possible because of strong misalignment or a poor signal-to-noise ratio, no diameter was measured. Scatter plots for NPC diameter analysis were created using Prism 9 (GraphPad).

## Statistics and reproducibility

Representative micrographs are provided in Fig. 1a and Extended Data Fig. 6a, f. The micrograph in Fig. 1a highlights the three-ring NPC architecture directly visualized in cryo-ET and was chosen from the dataset of 54 wild-type DLD-1 cell tomograms, all of which reproducibly show a three-ring architecture. The micrograph in Extended Data Fig. 6a was chosen from the auxin-depleted DLD-1 cell dataset (71 tomograms), and this image was specifically chosen because it highlights single-ring NPCs that we describe in this Article. Finally, the micrograph in Extended Data Fig. 6f was specifically chosen to provide the reader with an anecdotal observation that we identified only twice in the dataset (2 out of 73 tomograms) and these tomograms were excluded from subsequent processing (71 tomograms were used for subtomogram averaging).

## Reporting summary

Further information on research design is available in the Nature Research Reporting Summary linked to this paper.

## Data availability

Cryo-EM maps for the human DLD-1 NPC have been deposited in the EMDB with the following accession codes: EMD-12811 (CR), EMD-12812 (IR), EMD-12813 (NR) and EMD-12814 (full composite NPC). Coordinate files for the CR, IR and NR docked complexes have been deposited in the PDB with the following accession codes: 7PEQ (CR and NR complex) and 7PER (IR complex). Representative tilt series of lamella from DLD-1 cells have been deposited in the EMDB under accession code EMPIAR-10700 (wild-type, non-depleted cells) and EMPIAR-10701 (Nup96-depleted cells). Density maps for the human NPC from isolated nuclei used for comparison in this study can be found at the EMDB with accession number EMD-3103 (ref. [15]), as well as the recent NPC from HIV-infected T cells with accession number EMD-11967 (ref. [22]). Moreover, models

for the NR and CR Y complexes that we used as templates can be found in the PDB with accession code 5A9Q (ref. [15]), as well as the IR complex model with accession code 5IJN (ref. [9]).

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

**Acknowledgements** We thank the members of the Schwartz, Medalia and Weis laboratories for their discussions throughout this work; and especially M. Eibauer for help with data processing and providing scripts; and staff at the Arnold and Mabel Beckman Foundation for funding to develop the MIT.nano cryo-EM centre. This work was supported by the National Institutes of Health (nos R01-GM77537 and R35-GM141834, to T.U.S.), the Swiss National Science Foundation (nos SNSF 31003A_179418, to O.M.; and SNSF 31003A_179275, to K.W.) and the Helen Hay Whitney Foundation (to A.P.S.).

**Author contributions** A.P.S. and T.U.S. conceived the study. A.P.S. prepared human cryo-FIB samples and acquired the tomography data. M.W. calculated the structures and analysed the data. D.M. and A.K.R.L.-J. assisted in cryo-FIB sample preparation on the Zeiss Crossbeam 540. P.V.D. and E.J.B. helped in cryo-FIB preparation on the Thermo Fisher Scientific Aquilos and cryo-ET data acquisition and processing. M.T. and R.K.-T. prepared MEF samples for diameter analysis. S.R. and M.D. generated the *NUP96::Neon-AID* DLD-1 cell line. A.P.S., M.W, O.M. and T.U.S. wrote the manuscript with input from M.D. Funding was acquired by K.W., O.M. and T.U.S.

**Competing interests** The authors declare no competing interests.

### Additional information

**Correspondence and requests for materials** should be addressed to Ohad Medalia or Thomas U. Schwartz.

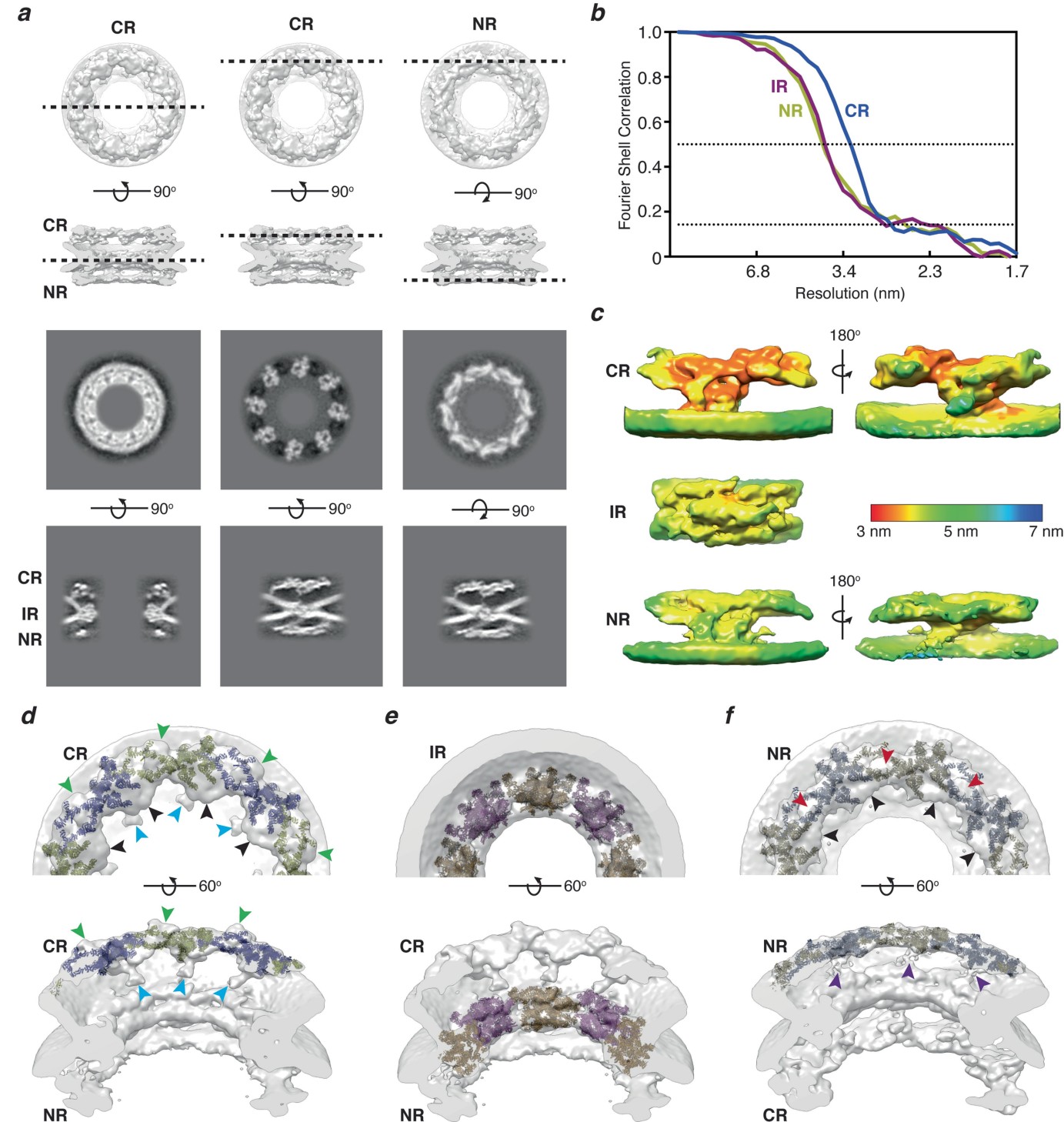

**Extended Data Fig. 1 | Details of *in situ* human NPC architecture. a**, Orthoslices through the nucleocytoplasmic axis, CR, and NR for our cryo-ET map of the human NPC from FIB-milled DLD-1 cells. Slice thickness: 3.4 nm. **b**, Fourier shell correlation curves of the CR, IR, and NR regions in our map. $FSC_{0.5}$ and $FSC_{0.143}$ are indicated as dotted lines. **c**, The local resolution for the CR, IR, and NR protomers is visualized as surface color. **d**, Segment of the CR to highlight fitting of the Y complex structure and several unique features including putative Nup358 density (green arrowheads), extra unassigned

density attached to the inner ring of the Y complex Nup85 (black arrowheads), and density we attribute to the Nup214 complex (blue arrowheads). **e**, Segment of the IR from our cryo-ET density with updated model fit inside. **f**, Segment of the NR to highlight fitting of the Y complex structure and several unique features including linker density between the Y complex rings (red arrowheads), unassigned density attached to the inner ring of the Y complex Nup85 (black arrowheads), and linkers between the Y complex inner ring and the IR (purple arrowheads).

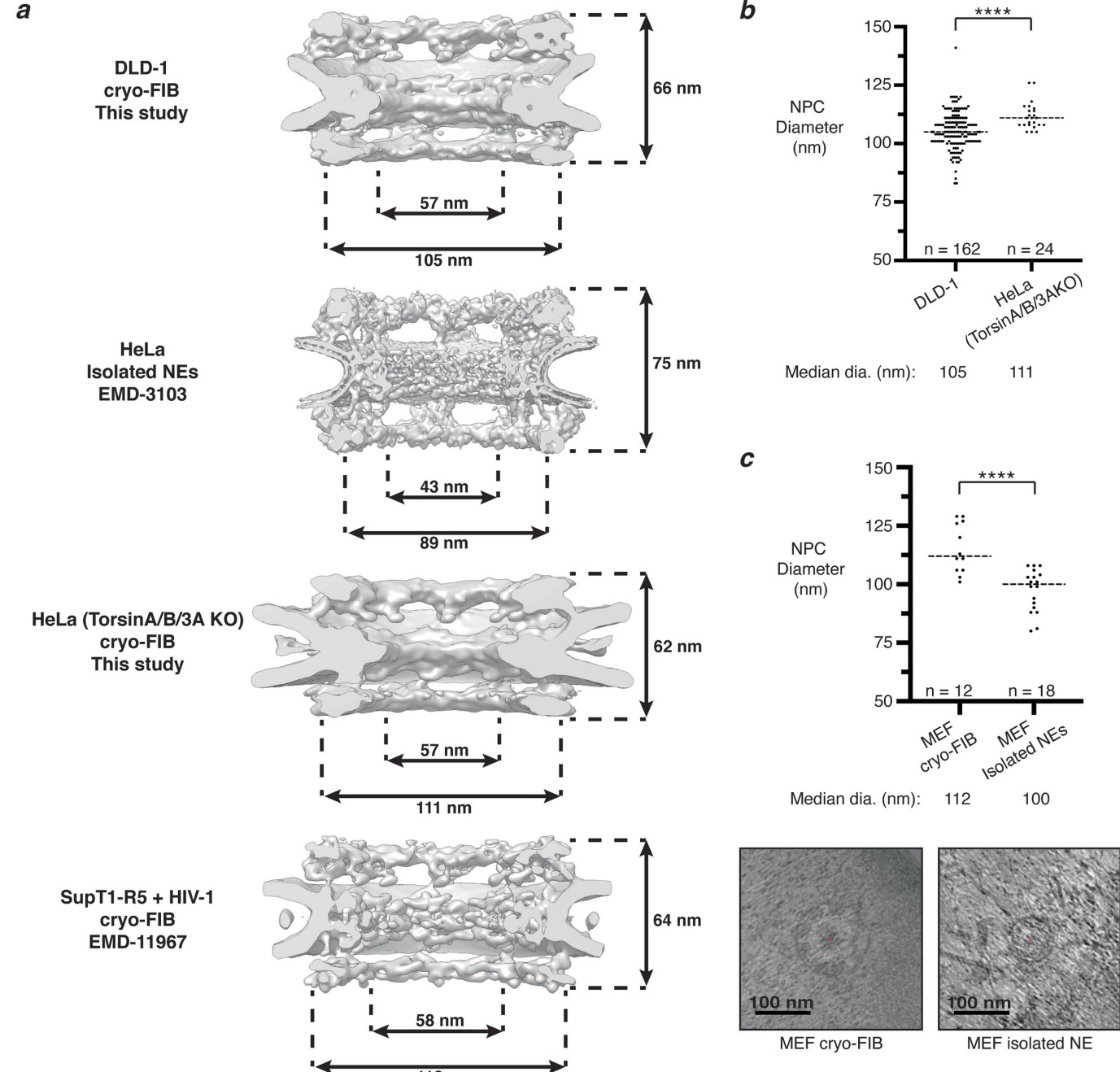

**Extended Data Fig. 2 | Comparison of NPC structures from purified NEs and cryo-FIB suggests the cellular environment impacts NPC architecture and IR diameter. a**, Comparison of human NPC density maps from cryo-FIB and nuclear envelope purification techniques. Measurements for the membrane-to-membrane and IR central channel diameters, as well as the height across the NE are shown. Interestingly, an asymmetric membrane curvature may also exist at human NPCs in HIV-infected cells[22], though the angles of the membranes appear different than ours. We refrain from making any speculations about this asymmetry because the deposited map comes from cells infected with HIV-1 virus. **b**, Measurements of NPC diameter from

single NPCs in DLD-1 cells (non-auxin treated) as well as HeLa cells (TorsinA/B/3A KO), both prepared by cryo-FIB. Median values from each distribution are indicated as a dotted line and below the plot. **** indicates a two-tailed p-value ≤ 0.0001 from Mann-Whitney U analysis. **c**, Measurements of NPC diameter from single NPCs in MEFs using cryo-FIB or nuclear envelope purification techniques. Median values from each distribution are indicated as a dotted line and below the plot. Representative tomographic slices from intact cells using cryo-FIB milling and from isolated NE preparations are shown. Slice thickness: 8.9 nm.

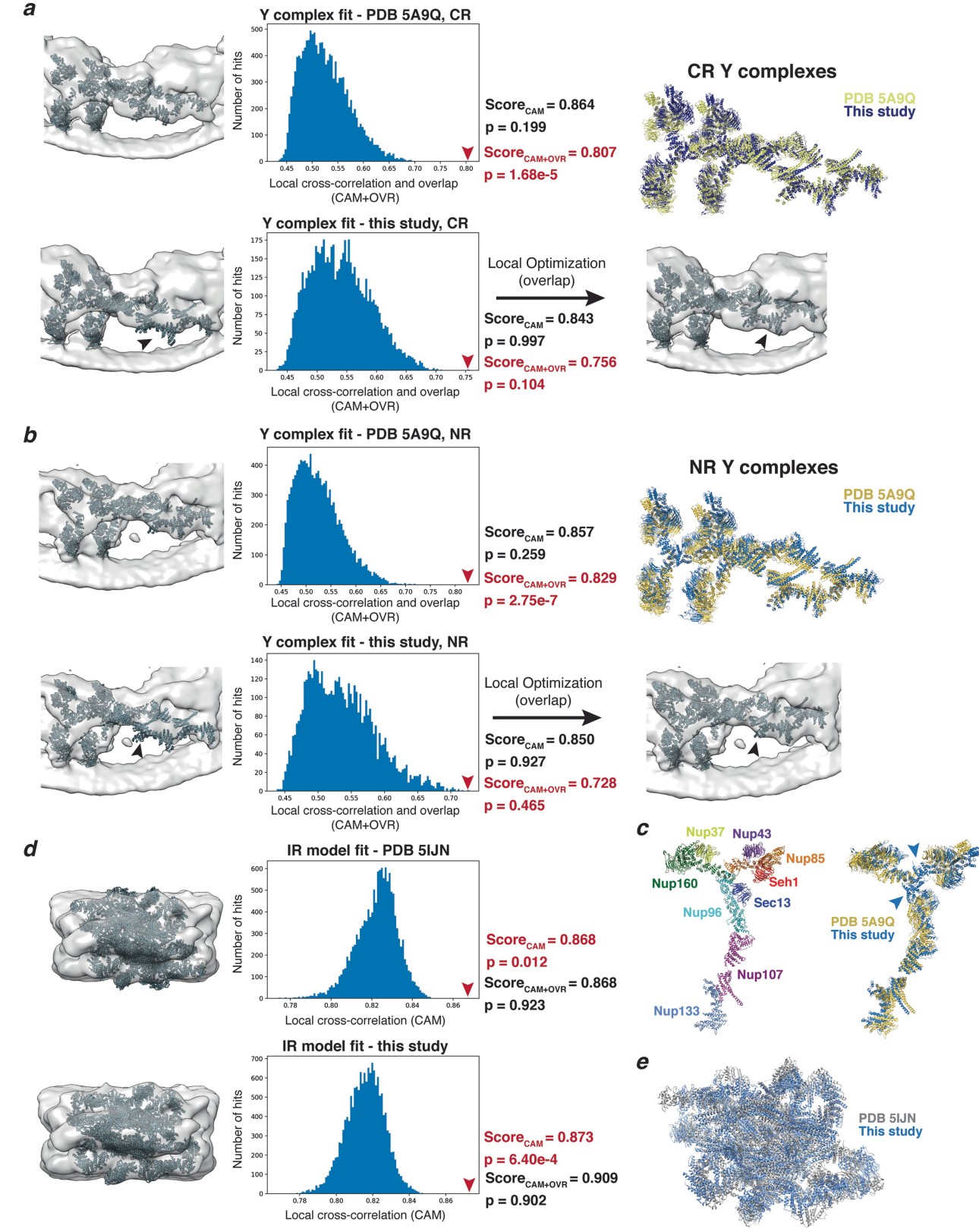

**Extended Data Fig. 3** | See next page for caption.

**Extended Data Fig. 3 | Systematic fitting of Y complex and Inner Ring complex into the human DLD-1 map. a**, Fitting of the Cytoplasmic Ring Y complex with the previous NPC model (PDB: 5A9Q) compared with our composite model. The details of this fitting are provided in the Methods section. All fitting runs were performed using Chimera with 1,000,000 random initial placements and local cross-correlation (Chimera's correlation about the mean, CAM), or a combination of local cross-correlation and overlap (CAM+OVR) for scoring. For each model, the top scoring model is shown to the left of a histogram of raw scores. The top scoring model is indicated on the histogram (red arrowhead) with its associated adjusted p-value for two different scoring metrics tested (red text represents the fit identified in the histogram, black represents the alternate metric). Two-sided P-values were corrected for multiple testing using the Benjamini-Hochberg procedure. The resulting composite Y model underwent an additional local optimization step to further improve the density fit (black arrowheads indicate regions that are improved with this step). An overlay of our model and the previous model is also shown in the panel. **b**, Same as **a**, but for the Nuclear Ring Y complex, also starting with PDB 5A9Q. **c**, Overlay of our composite Y complex model with that of the previous model. The additional modeled regions in our Y complex model "hub" (Nup160, 85, 96 and Sec13) and are indicated (blue arrowheads). **d**, Fitting of the Inner Ring complex with the previous IR model (PDB: 5IJN) compared with our model. **e**, Overlay of our modified IR complex model with the previous model.

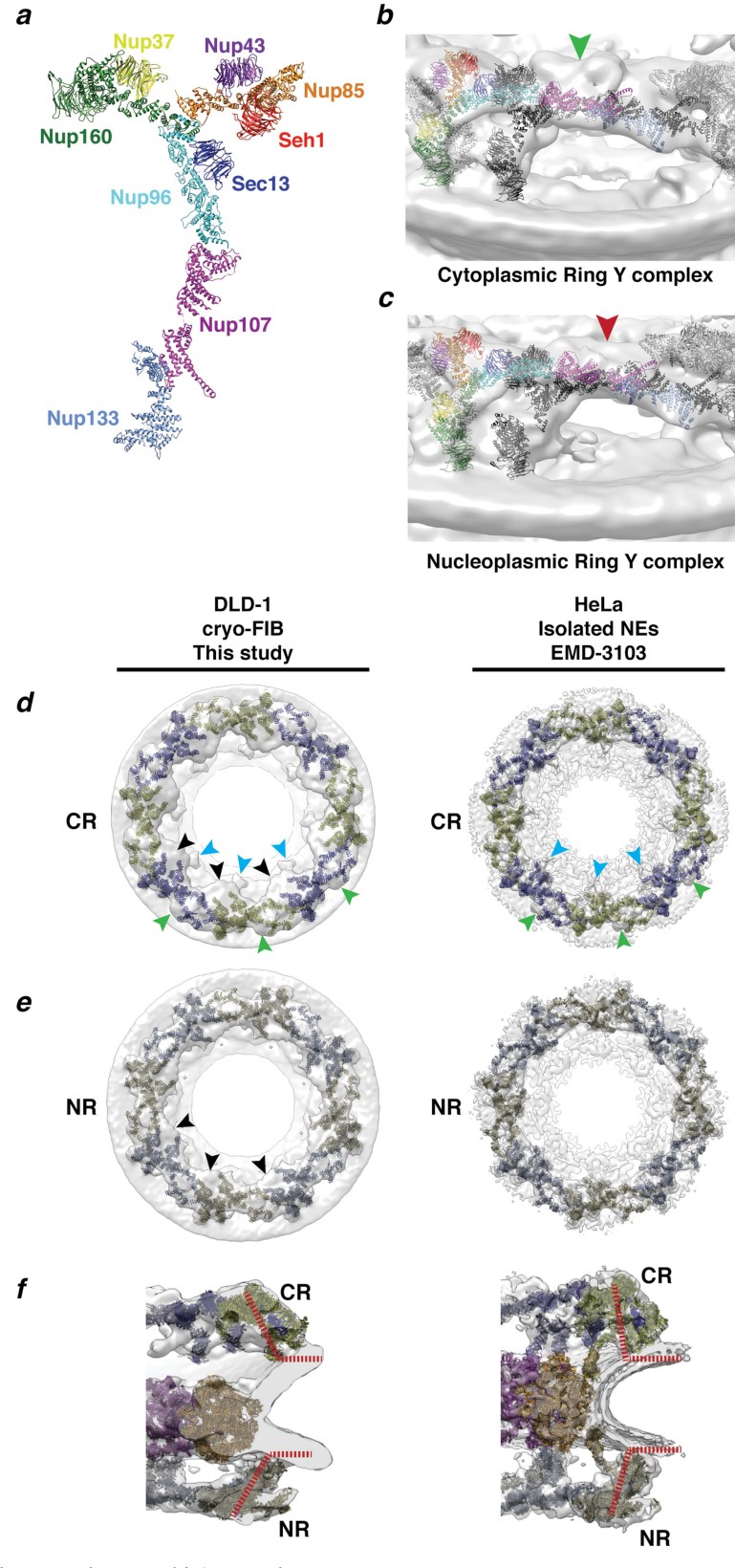

**Extended Data Fig. 4 | Details of the Y complex assembly in CR and NR.**
**a**, Cartoon model for the human composite Y complex that was docked into the CR of our cryo-ET density map. **b**, Zoom-in view of the docked Y complex model in the CR with additional density connecting the two Y rings attributed to the Nup358 complex (green arrowhead). **c**, Zoom-in view of the docked Y complex model in the NR with additional density connecting the two Y rings (red arrowhead). **d**, Comparison of CR sub-structures from our study and the previous human model. Arrowheads indicate density attributed to the Nup358 complex (green), Nup214 complex (blue), and an unassigned density we observe at the Nup85 arms (black). **e**, Same as (**d**), but for NR. Black arrowheads indicate similar density at Nup85 arms as in CR. **f**, Comparison of cross-section views of the NPC with dashed red line to indicate the orientation of the Y complexes with respect to the nuclear membranes.

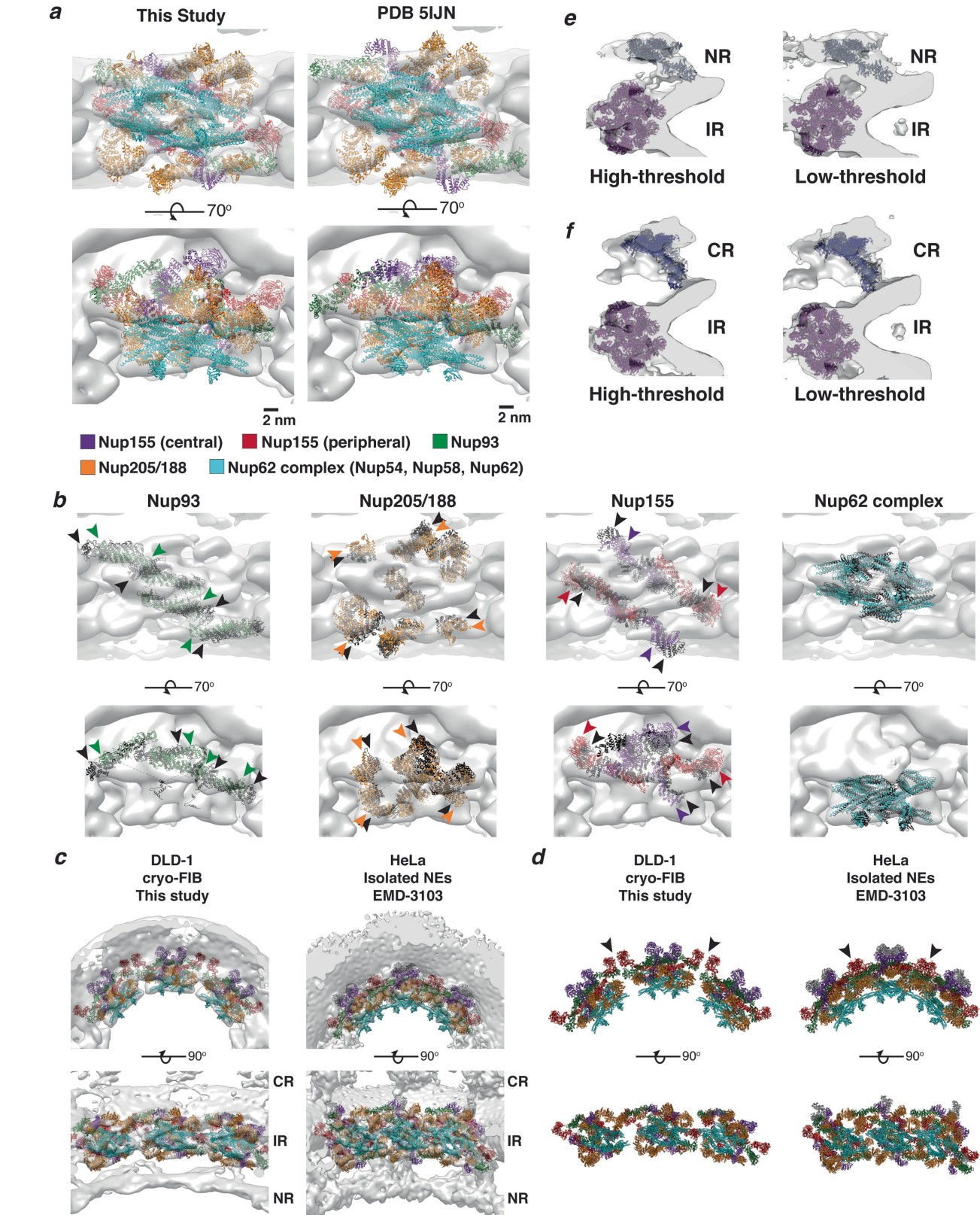

**Extended Data Fig. 5** | See next page for caption.

**Extended Data Fig. 5 | Updated model for native IR. a**, Side-by-side comparison of the previous IR model (PDB: 5IJN) with our updated model to fit the DLD-1 NPC map. **b**, Superimposition of our updated IR model with the previous model shows the rearrangements required to fit our density map. Each pane highlights a different nucleoporin colored in our updated model as in (**a**), or colored dark grey from the previous model. **c**, Views of three updated IR-models in our map compared with the previous model and human NPC map. **d**, Same as (**c**) but with only models. **e**, Cross-section views of the NR in our map at both high and low threshold to show connectivity with the IR. **f**, Same as (**e**), though no density can be observed between the CR-IR, likely due to flexibility. Flexibility could explain why the ring-connecting densities, proposed to be Nup155[9], remain conspicuously poorly resolved in our map compared to the previous human maps[12,15,16]. We hypothesize that this observation may be unique to the more "constricted" IR state observed previously, locking the ring-connecting proteins into a more rigid state. In the same vein, others have shown that NPCs from yeast, both semi-purified[36] and in their native state[18], can contain multiple flexible linkers between the outer rings and IR, suggesting this arrangement could provide resilience to the NPC structure.

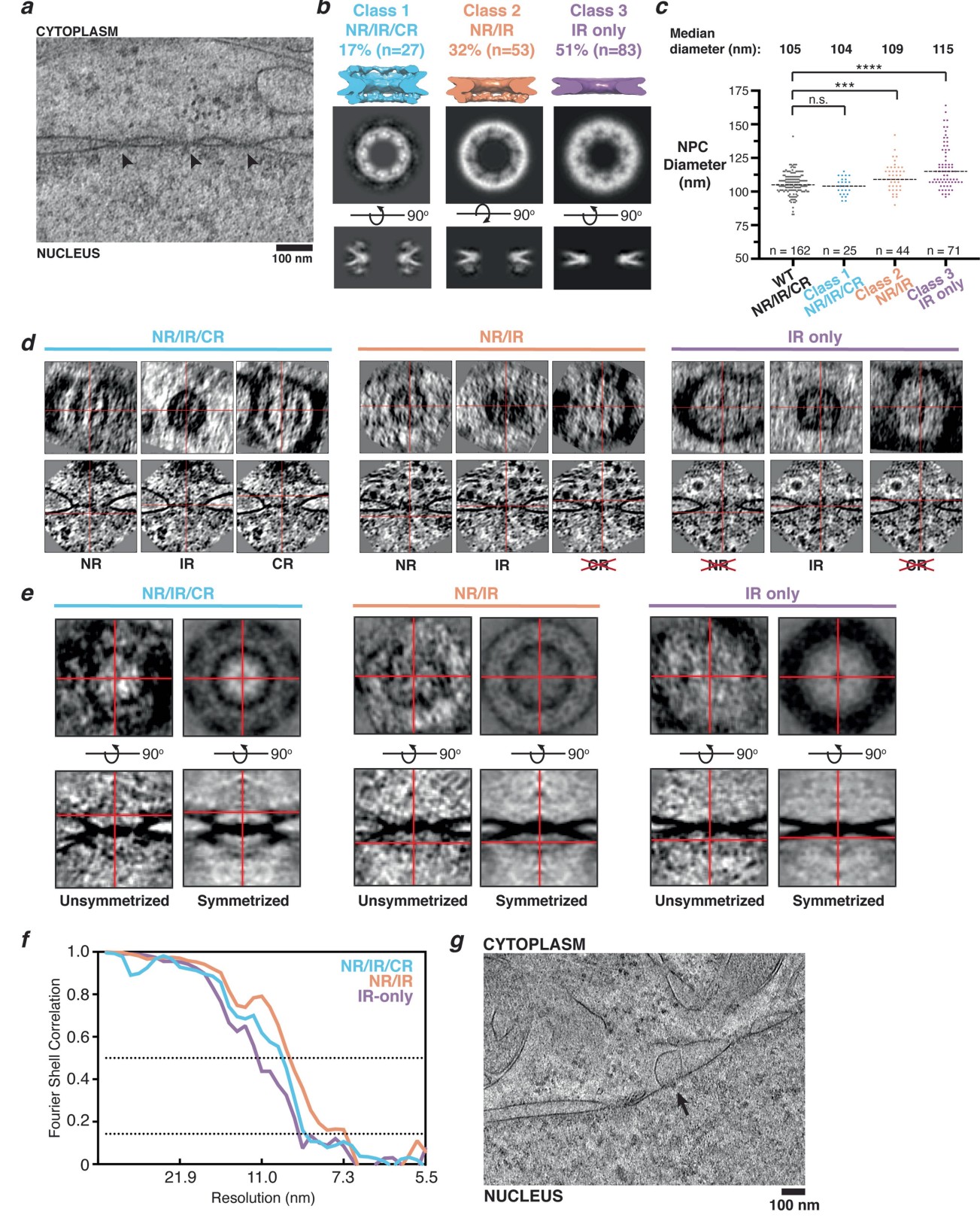

**Extended Data Fig. 6** | See next page for caption.

**Extended Data Fig. 6 | Details of Nup96-depleted NPC structures.**
**a**, Tomographic slice from a lamella prepared by cryo-FIB of NUP96::Neon-AID DLD-1 cells in the presence of auxin for 4 h (Nup96-depleted). Arrowheads indicate Nup96-depleted NPCs. **b**, Orthoslices across the nucleocytoplasmic axis for the final aligned maps of the three Nup96-depleted NPC classes. **c**, Measurements of NPC diameter from single NPCs in each of the data sets as well as the non-auxin treated (WT). *** indicates a two-tailed p-value ≤ 0.001 and **** indicates a two-tailed p-value ≤ 0.0001 from Mann-Whitney U analysis of each distribution compared with the wild-type condition. Median values from each distribution are indicated as a dotted line and shown above the plot. **d**, Orthoslices across the nucleocytoplasmic axis of single NPCs in each of the three Nup96-depleted NPC classes. Slice thickness is 4.1 nm. **e**, Orthoslices across the nucleocytoplasmic axis of class averages after manual orientation alignment and classification. **f**, Fourier shell correlation curves for the Nup96-depleted NPC classes. $FSC_{0.5}$ and $FSC_{0.143}$ are indicated as dotted lines. **g**, Tomographic slice of Nup96-depleted cells showed instances of NE herniation at the ONM (example indicated by arrow). Slice thickness for all panels except d: 2.7 nm.

## Extended Data Table 1 | Summary of cryo-ET imaging parameters

| | Wild-type (No Auxin) cells | Nup96-depleted (+Auxin) cells | HeLa (TorsinA/B/3A knockout) cells |
|---|---|---|---|
| Magnification | 26,000x | 26,000x | 26,000x |
| Voltage (keV) | 300 | 300 | 300 |
| Detector | Gatan K3 with Bioquantum | Gatan K3 with Bioquantum | Gatan K3 with Bioquantum |
| Energy filter | Yes | Yes | Yes |
| Slit width (eV) | 20 | 20 | 20 |
| Pixel size (Å) | 3.4 | 3.4 | 3.4 |
| Defocus range (µm) | 2.5 to 5.0 | 2.5 to 5.0 | 5.0 |
| Defocus step (µm) | 0.5 | 0.5 | N/A |
| Tilt range (min/max, step | -52°/+68°, 2° | -52°/+68°, 2° | -52°/+68°, 2° |
| Tilt scheme | Dose-symmetric (Hagen scheme) | Dose-symmetric (Hagen scheme) | Dose-symmetric (Hagen scheme) |
| Total dose (e$^-$/Å$^2$) | ~145 | ~145 | ~120 |
| Tomograms used | 194 | 163 | 27 |

Imaging parameters for tilt series acquisition contributing to each dataset are provided in the table.

# Reporting Summary

## Statistics

For all statistical analyses, confirm that the following items are present in the figure legend, table legend, main text, or Methods section.

| n/a | Confirmed | |
|---|---|---|
| ☐ | ☒ | The exact sample size ($n$) for each experimental group/condition, given as a discrete number and unit of measurement |
| ☐ | ☒ | A statement on whether measurements were taken from distinct samples or whether the same sample was measured repeatedly |
| ☐ | ☒ | The statistical test(s) used AND whether they are one- or two-sided<br>*Only common tests should be described solely by name; describe more complex techniques in the Methods section.* |
| ☒ | ☐ | A description of all covariates tested |
| ☒ | ☐ | A description of any assumptions or corrections, such as tests of normality and adjustment for multiple comparisons |
| ☒ | ☐ | A full description of the statistical parameters including central tendency (e.g. means) or other basic estimates (e.g. regression coefficient) AND variation (e.g. standard deviation) or associated estimates of uncertainty (e.g. confidence intervals) |
| ☐ | ☒ | For null hypothesis testing, the test statistic (e.g. $F$, $t$, $r$) with confidence intervals, effect sizes, degrees of freedom and $P$ value noted<br>*Give P values as exact values whenever suitable.* |
| ☒ | ☐ | For Bayesian analysis, information on the choice of priors and Markov chain Monte Carlo settings |
| ☒ | ☐ | For hierarchical and complex designs, identification of the appropriate level for tests and full reporting of outcomes |
| ☒ | ☐ | Estimates of effect sizes (e.g. Cohen's $d$, Pearson's $r$), indicating how they were calculated |

*Our web collection on statistics for biologists contains articles on many of the points above.*

## Software and code

Policy information about availability of computer code

| | |
|---|---|
| Data collection | Tomo v5.3.0 (ThermoFisher) for operation of electron microscope (referenced in Methods) |
| Data analysis | Cryo-ET data processing (all available and referenced in Methods): IMOD v4.11.4 (open source, cited), TOM toolbox (open source, cited), Relion v3.1.2_cu9.2 (open source, cited), MATLAB R2019a (MathWorks)<br>Modeling/visualization: Chimera v1.15rc (open source, cited), IMOD v4.11.4 (open source, cited), TOM toolbox v2008 (open source, cited)<br>Statistics/data visualization: Python v3.8.8, Matplotlib v3.3.4, SciPy v1.6.2, StatsModels v0.12.2, Prism v9 (Graphpad) |

For manuscripts utilizing custom algorithms or software that are central to the research but not yet described in published literature, software must be made available to editors and reviewers. We strongly encourage code deposition in a community repository (e.g. GitHub). See the Nature Portfolio guidelines for submitting code & software for further information.

## Data

Policy information about availability of data

All manuscripts must include a data availability statement. This statement should provide the following information, where applicable:

- Accession codes, unique identifiers, or web links for publicly available datasets
- A description of any restrictions on data availability
- For clinical datasets or third party data, please ensure that the statement adheres to our policy

Cryo-EM maps for the human DLD-1 NPC have been deposited in the Electron Microscopy Data Bank (EMDB) with the following accession codes: EMD-12811 (CR), EMD-12812 (IR), EMD-12813 (NR), and EMD-12814 (full composite NPC). Coordinate files for the CR, IR, and NR docked complexes have been deposited in the Protein Data Bank (PDB) with the following accession codes: PDB-7PEQ (CR and NR complex) and PDB-7PER (IR complex). Representative tilt series of lamella from DLD-1 cells have been deposited in the EMDB under accession code EMPIAR-10700 (wild-type, non-depleted cells) and EMPIAR-10701 (Nup96-depleted cells).

# Field-specific reporting

Please select the one below that is the best fit for your research. If you are not sure, read the appropriate sections before making your selection.

☒ Life sciences ☐ Behavioural & social sciences ☐ Ecological, evolutionary & environmental sciences

For a reference copy of the document with all sections, see nature.com/documents/nr-reporting-summary-flat.pdf

# Life sciences study design

All studies must disclose on these points even when the disclosure is negative.

| | |
|---|---|
| Sample size | Sample sizes were determined by available cryo-electron microscopy and cryo-FIB instrument time. For the wild-type NPC structure, 194 NPCs were extracted from 54 tomograms. For Nup96-depleted NPC structures, 163 NPCs were extracted from 71 tomograms. The sample size is sufficient to obtain a structure at the reported resolution. |
| Data exclusions | For cryo-ET: Tomograms exhibiting errors during data collection or lacking NPCs were excluded. During subtomogram averaging and alignment some incomplete NPC subvolumes were manually removed (described in Methods). Exclusion of error-containing or incomplete data is a standard, pre-established practice for cryo-ET processing.<br><br>For NPC diameter measurements: When an accurate measurement was not possible because of strong misalignment or poor signal to noise, no diameter was measured. Sample size for diameter measurements is included with each plot. |
| Replication | Tomograms were acquired from multiple cell samples in each condition and thus served as biological replicates. We have included a statement of reproducibility in regards to the representative micrographs in the manuscript. Representative micrographs are provided in Fig. 1a and Extended Figs. 6a, 6f. The micrograph in Fig. 1a highlights the 3-ring NPC architecture directly visualized in cryo-ET and was chosen from the dataset of 54 wild-type DLD-1 cell tomograms, all of which reproducibly show a 3-ring architecture. The micrograph in Extended Data Fig. 6a was chosen from the auxin-depleted DLD-1 cell dataset (71 tomograms), and this image specifically chosen because it highlights single-ring NPCs that we describe in this manuscript. Finally, the micrograph in Extended Data Fig. 6f was specifically chosen to provide the reader with an anecdotal observation we only identified twice in the dataset (2 out of 73 tomograms) and these tomograms were excluded from subsequent processing (71 tomograms used for subtomogram averaging). |
| Randomization | Cells for cryo-FIB milling and tomogram collection were chosen at random on each TEM grid. Division of the two half-sets of data used for resolution estimation was based on the tomogram number being even or odd. |
| Blinding | Blinding was performed during sample preparation and data collection as cellular samples were given a unique identification number and only de-identified afterward for classification and data processing procedures. |

# Reporting for specific materials, systems and methods

We require information from authors about some types of materials, experimental systems and methods used in many studies. Here, indicate whether each material, system or method listed is relevant to your study. If you are not sure if a list item applies to your research, read the appropriate section before selecting a response.

## Materials & experimental systems

| n/a | Involved in the study |
|---|---|
| ☒ | ☐ Antibodies |
| ☐ | ☒ Eukaryotic cell lines |
| ☒ | ☐ Palaeontology and archaeology |
| ☒ | ☐ Animals and other organisms |
| ☒ | ☐ Human research participants |
| ☒ | ☐ Clinical data |
| ☒ | ☐ Dual use research of concern |

## Methods

| n/a | Involved in the study |
|---|---|
| ☒ | ☐ ChIP-seq |
| ☒ | ☐ Flow cytometry |
| ☒ | ☐ MRI-based neuroimaging |

## Eukaryotic cell lines

Policy information about cell lines

| | |
|---|---|
| Cell line source(s) | Human colorectal adenocarcinoma cells (DLD-1) with homozygous insertion at Nup96 loci containing NeonGreen moiety and an auxin-inducible degron (Nup96::Neon-AID) from laboratory of M. Dasso (Ref. 23: Regmi et al., 2020). HeLa-derived cell line was created and authenticated in the laboratory of T. Schwartz in an earlier study (Ref. 43: Demircioglu et al., 2019). |
| Authentication | Creation and authentication of the Nup96::Neon-AID DLD-1 cell line was performed in a separate study (Ref. 23: Regmi et al., 2020). Briefly, PCR assays followed by sequencing of PCR products confirmed homozygous insertion of the Neon-AID tag. Subsequently, Western blot analysis confirmed a larger MW protein due to the epitope tag. The HeLa-derived cell line was |

authenticated in a previous study (Ref 43: Demircioglu et al., 2019).

Mycoplasma contamination

Cell lines were not tested for Mycoplasma contamination.

Commonly misidentified lines
(See ICLAC register)

No commonly misidentified cell lines were used in this study.

