## [Peer Review File · Nature]

Manuscript Title: The cellular environment shapes the nuclear pore complex architecture

Reviewer Comments & Author Rebuttals

Reviewer Reports on the Initial Version:

Referee #1 (Remarks to the Author):

In this study, the authors use FIB milling and cryo-ET to examine the structure of the in situ human NPC and the effects on it of depletion of a major outer ring component. The structure of NUP96::Neon-AID labeled human NPCs was mapped by FIB-cryo-ET first, and showed that – by comparison with previous studies on isolated human and other vertebrate NPCs, but in agreement both with previous studies on human NPC and comparisons of yeast isolated versus in situ NPCs, significantly larger inner ring diameters are observed in situ. Moreover, as with Mahamid et al., 2016 (PMID: 26917770), they observe that individual NPCs in situ appear to come in a wide range of inner ring diameters. They also observe an asymmetric curvature of the NE membrane in the nucleocytoplasmic axis, which (also consistent with prior studies in yeast) the authors point out is likely conferred by the NPC ring substructures. While others have studied both in situ and isolated NPCs in vertebrates, explicit comparisons were not made in as much detail as here.

To further analyze the interdependence of the different NPC rings, the authors compare their initial map with the ones obtained using a similar approach upon auxin-induced degradation of the outer ring component Nup96 (derived by autoproteolytic cleavage of a 186 kDa precursor into Nup98 and Nup96), which forms the central stalk of the Y-shaped Nup107-160 complex comprising outer rings. The effects of Nup96 degradation, compared with that of degradation of other outer ring nups (Nup107 and Nup160) and two inner ring Nups (Nup188 and Nup93) were previously examined in an extremely elegant Biorxiv preprint by some of the same authors (Regmi et al., 2020), that comprehensively employed fluorescence microscopy, proteomics and scanning electron microscopy to convincingly demonstrate a surprising degree of structural independence between the nuclear and cytoplasmic outer rings and the inner ring in mammalian cells.

In the present study, the authors show that upon Nup96 depletion, several NPC forms could be detected, basically differing in the absence of the cytoplasmic and nuclear rings while the inner ring remained essentially unaffected. It is not clear though if the observed Nup96 degradation-induced NPC forms are mature NPCs getting disassembled or dead-ends of failed newly assembled NPCs. However they strongly indicate that the C and N outer rings may restrict the inner ring dimensions, intriguingly acting as molecular “rulers” to define the upper limit of the inner ring’s size, as well as modulate NE membrane curvature.

Overall this is a well performed study with a number of intriguing findings, but rather limited in scope, with many of the conclusions – though given in more detail – being ones already covered by others; in essence, this manuscript revisits the in situ FIB milling cryo-ET of human cells first comprehensively surveyed by Wolfgang Baumeister’s group in Mahamid et al., and to some extent by Martin Beck’s lab in Zila et al, 2021 (PMID: 33571428). In combination with the Regmi et al. studies the two manuscripts collectively tell a very compelling story, but (though this is a decision for the editor) this work alone may be too limited in scope for this Journal. There are also a couple of point that the authors could address:

- There are no metrics reported in the manuscript that could help assess the level of confidence of the models generated by the authors for the ORs and IR. The average resolution of the cryo-ET maps is modest (~3.4 nm), and, in principle, not detailed enough to allow an accurate positioning

of the different components of the NPC. Known NPC features are apparently missing from the map density, like the Nup155 units connecting the IR and cytoplasmic OR; is this due to their flexibility, or to heterogeneity, or the type of cell line used in the study? The authors specifically mention the papers from Ori et al., 2013 and Raices et al., 2012 where changes in Nup stoichiometry are detected between mammalian cell lines. However, they choose to assume a certain composition and stoichiometry for the strain they used here – but do they know that this cell line indeed has the same stoichiometry and composition they assume? Taking all these considerations together, the comparison of Nup and subcomplex positions between NPC states (in situ vs isolated NEs) remains a bit speculative and potentially ambiguous.

- It has been shown that in baker's yeast the ORs are only connected through a few flexible linkers to the IR (Kim et al., 2018; Allegretti et al., 2020), and it was suggested that such an arrangement provides flexibility and resilience to the NPC. It would be interesting to discuss and compare those observations with the observations made by the authors in the human in situ NPC. Previous vertebrate NPC maps suggested a much more solid connection between rings.

Referee #2 (Remarks to the Author):

This manuscript by Schuller et al. uses focused ion beam (FIB) milling followed by cryo-electron tomography (cryo-ET) and subtomogram averaging to resolve a native structure of the human nuclear pore complex (NPC) within DLD-1 colon cancer cells. Compared to previous cryo-ET structures from isolated nuclei, the authors observe that the NPC's inner ring is significantly dilated, and gaps are present between the inner ring's asymmetric subunits (also called "spokes"). Overall, the NPC has a shorter structure along its nucleo-cytoplasmic axis, with the outer ring Y-complexes angled toward the central channel. The authors also observe asymmetry in the nuclear envelope, which they infer may be caused by asymmetric forces from the two outer rings. Finally, they used cryo-ET to generate low-resolution averages of NPCs in various states of disassembly following Nup96-depletion, which was previously shown to disassociate the outer rings from the NPC. From this depletion experiment, they propose that the nuclear ring is more firmly attached to the NPC than the cytoplasmic ring, and that the outer rings limit the maximum dilation of the inner ring.

We have a mixed opinion of this manuscript. On the positive side, this appears to be the best-resolved in situ human NPC structure to date. While the estimated resolution of the subtomogram averaging (34 Å) is not a dramatic improvement on a previously-published in situ human NPC (37 Å) (Zila et al., 2021; Cell), the densities in the map certainly appear to be more defined. As the authors acknowledge (lines 248-249), the moderate resolution of their structure is insufficient to model molecular structures into significant portions of their NPC density map (only the Y-complexes are modeled in the cytoplasmic and nuclear rings). Thus, future improvements will be required to produce a more "definitive" human NPC structure. However, the NPC is a challenging target due to its size and flexibility, and the success of this study should not be undervalued-- the structure presented in this manuscript is the current benchmark for native NPC architecture in human cells.

NOVELTY:

The biggest issue with this manuscript has to do with how it is presented. Throughout the abstract, introduction, and results, the authors assert the novelty of the paper's main finding: the NPC's inner ring is more dilated inside the cell than in isolated nuclei. Only in the discussion section do the authors begin to reveal that this has been well documented before (lines 187-190: "The ability to modulate its diameter may be a general property of NPCs as we observed this phenomenon in mouse embryonic fibroblasts, and it has also recently been reported in *S. cerevisiae* [34] and in various growth conditions for *S. pombe* [39]"; lines 212-214: "A recent study of HIV infected T

cells is consistent with such IR flexibility and revealed a similarly widened NPC channel, large enough to accommodate the intact HIV-1 capsid [42]).

To the best of our knowledge, there are FOUR previous studies that have resolved in situ structures of NPCs inside cells using a combination of FIB milling and cryo-ET. These studies all describe the asymmetry of the NPC and dilation of the highly-conserved inner ring. Their results should be presented in the introduction so the reader can understand the current state of knowledge in the field and how Schuller et al. adds to it. We list these studies below:

1) Mosalaganti et al., 2018; Nature Comm; <https://doi.org/10.1038/s41467-018-04739-y>
This study presents the native structure of the *C. reinhardtii* NPC (resolution: 30 Å) and provides an in-depth comparison to the isolated human NPC structure (with figures that are quite similar to Fig. 3 here in Schuller et al.). The dilation of the inner ring and resulting “peripheral channel” spaces between the spokes are shown in detail. The reduced NPC height vs. the isolated human NPC is also clearly documented. The authors propose that this is a species-specific difference. However, as Schuller et al. observe reduced height of the human NPC in situ, this algal NPC result could be reconsidered in the discussion.

Key descriptions:

“The inner ring of the CrNPC map is dilated in comparison to the HsNPC map, with substantial spacing between its rotationally symmetric spokes, thereby forming relatively large peripheral channels that have been proposed to accommodate the import of inner nuclear membrane proteins.”

“Are these species-specific differences in the inner ring, or could they be related to the NPC’s functional state? ...We speculate that not only the FG-rich regions, but also the scaffold of the NPC may be much more dynamic than anticipated.”

2) Allegretti et al., 2020; Nature; <https://doi.org/10.1038/s41586-020-2670-5>
This study presents a well-resolved native structure of the *C. cerevisiae* NPC (resolution: 25 Å). The dilated inner ring and resulting spaces between the spokes are very clearly shown.

Key description:

“The entire IR has to be dilated by about 20 nm in diameter, thereby spatially separating the eight individual spokes. This analysis underlines the plasticity of the NPC within cells, which might be physiologically relevant for the transport of large cargos and inner nuclear membrane proteins.”

3) Zimmerli et al., 2020; bioRxiv; <https://doi.org/10.1101/2020.07.30.228585>
This study presents a well-resolved native structure of the *C. pombe* NPC (resolution: 23 Å). We consider this to be the current definitive paper on inner ring dilation, as it demonstrates how the inner ring dilates and contracts in response to the cell’s energy state. The spaces between the inner ring spokes are also very clear in the dilated NPC structure.

Key descriptions:

“The central channel constriction of the IR is more elaborate and mediated by a lateral displacement of the 8 spokes that move as independent entities to constrict or dilate the IR. In the dilated state, around 3-4 nm wide gaps are formed in between the neighboring spokes, while in the constricted state the spokes form extensive contacts, equivalent to those in the previously published structures of the human NPC in isolated nuclear envelopes”.

“Our dynamic structural model suggests large scale conformational changes that occur by movements of the spokes with respect to each other but largely preserve the arrangement of

individual subcomplexes. Previous structural models obtained from isolated nuclear envelopes thereby represent the most constricted NPC state”.

4) Zila et al., 2021; Cell; <https://doi.org/10.1016/j.cell.2021.01.025>

This study presents the native structure of the human NPC (resolution: 37 Å) and describes in detail how the inner ring is dilated in both HIV-infected and non-infected cells. This confirms the conservation of inner ring dilation from algae and yeast to humans. The spaces between the inner ring spokes are visible in the subtomogram average (Fig. S7B). The reduced height of the NPC in situ is also apparent. The membrane curvature is asymmetric across the NPC (Fig. 6C), but the angles do look different than what is shown here in Schuller et al., which could be a point of discussion.

Key description:

“In line with other studies conducted in intact human cells (Beck and Baumeister, 2016; Mahamid et al., 2016), the NPCs appeared dilated in comparison to isolated nuclear envelopes and displayed an average diameter of ~64 nm. ...We found that the NPCs dilated to similar diameters in non-infected T cells. Taken together, our findings indicate that the NPC structure observed under the relevant conditions, namely in infected and non-infected T cells in situ, is representative of the transporting state, whereas the constricted state observed in isolated nuclear envelopes (von Appen et al., 2015) may be more relevant to stress conditions (Zimmerli et al., 2020).”

In light of the above studies, the following statements in Schuller et al. overstate their novelty and should be reworded to put their observations into context with the NPC field:

Abstract: “Our findings highlight the inherent flexibility of the NPC and strongly suggest that the cellular environment has a hitherto underappreciated influence on NPC dimensions and architecture.”

-> How can the in situ effect on NPC architecture possibly be “hitherto unappreciated”?

Lines 38-39: “Surprisingly, the organization of the IR increases the central pore diameter by more than 30%”

-> How is this a surprise? It’s to be expected based on previous studies.

Lines 43-44: “Our study shows that the cellular environment significantly influences the diameter of the central channel of the NPC”

-> We would say that it corroborates this previously-reported effect of the cellular environment.

Lines 183-185: “Our study reveals a substantial NPC plasticity and shows that purification of NPCs and the NE influences the NPC structure.”

-> Rather than “reveals”, we would say that this study strongly confirms the previous hypothesis (from algae, yeast, and human cells) that purification of NPCs and the NE influences the NPC structure.

Lines 211-216: “Furthermore, we observe that the subunits in the IR contain gaps between adjacent subunits, while previous models suggested a compact, interwoven architecture. ...This flexibility could be important for protein trafficking to the INM which requires passage through the NPC while membrane anchorage is retained.”

-> As we outline above, all four previous studies have observed “peripheral channel” gaps between the spokes of the dilated inner ring, and thus should be correctly cited. The idea that these gaps could permit passage of membrane proteins through the NPC has also been proposed:

...proposed in Mosalaganti et al. 2018: “The inner ring of the CrNPC map is dilated in comparison to the HsNPC map, with substantial spacing between its rotationally symmetric spokes, thereby

forming relatively large peripheral channels that have been proposed to accommodate the import of inner nuclear membrane proteins.”

...proposed in Allegretti et al., 2020: “This analysis underlines the plasticity of the NPC within cells, which might be physiologically relevant for the transport of large cargos and inner nuclear membrane proteins.”

...proposed in Zimmerli et al., 2020: “Peripheral channels are thought to be important for the nuclear import of inner nuclear membrane proteins. Here we observed around 3-4 nm wide lateral gaps between the individual spokes of actively transporting NPCs. Notably, our data processing workflow yields an average of conformation under the respective conditions and individual spokes are even more dynamic. Therefore, it is plausible that the opening and closing of peripheral channels may regulate the translocation of inner nuclear membrane proteins.”

Lines 216-222: “A wider central channel does not only have implications for the maximal size of soluble and membrane-bound cargo that can pass through the NPC, but it also changes the density of the FG-repeats that fill the transport channel. The potential of the FG density to influence transport has been documented in model systems that focused on this aspect of NPC biology. Our NPC model indicates that the FG-density in the central channel is lower than previously thought, or that flexibility, particularly within the IR, may be a means to modulate the properties and selectivity of the FG-barrier.”

-> This hypothesis of inner ring dilation changing FG concentration has also been proposed before, and should be cited as such:

...proposed in Mosalaganti et al. 2018: “The local FG-Nup concentration within the central channel might change during inner ring dilation.”

...proposed in Zimmerli et al., 2020: “The estimated volume of the central channel in the most dilated state was almost twice as large (~152'000 nm³) as compared to the most constricted state (~86'000 nm³), which likely translates to ~2-fold change in concentration of the FG-repeats contained therein.” “It therefore is plausible that a constricted central channel volume leads to an increased local FG-domain concentration which in turn limits the passive diffusion of molecules of a constant size.”

Lines 227-228: “Our data might indicate that the CR/NR act as molecular “rulers” to define the upper limit of the IR dimensions.”

-> Again, this hypothesis has been proposed before and should be cited:

...proposed in Mosalaganti et al. 2018: “The CrNPC inner ring has the same diameter as the outer rings, which are horizontally stacked upon it. In this conformation, the head-to-tail connection of the outer ring Y-complexes might be important for restricting the maximal dilation of the pore.”

In summary, the main findings in this paper build on previous work and must be presented in that context. The good news is that this can be relatively easily addressed by reworking the text of the manuscript. There is the perfect opportunity to discuss how the in situ architecture and dilation of the inner ring are conserved between species (algae, yeast, humans), while the outer rings vary widely. As for the question of “novelty”, while it is true that many of the main findings in this paper have been reported in some fashion before, we do not believe this should prohibit publication. The improved structure of the human NPC is an important advance and adds a new layer to our understanding of NPC architecture across the tree of life.

TECHNICAL ISSUES:

In addition, there are several technical issues found in the Methods section that should be

addressed:

Tilt series acquisition and processing:

"CTF was determined and corrected as described previously"

-> Such a crucial step should be described in the paper instead of pointing to an external reference. Was it done by phase flipping? CTF multiplication? Wiener filtering? Was it applied in 2D (defocus gradient on images of tilt series) or was it a 3D correction (e.g., NovaCTF)? Which software was used? (cite it!). These details should be given.

Subtomogram averaging:

"At this stage incomplete NPC sub-volumes were excluded based on manual inspection (1252 protomers)."

-> It is not clear here what was excluded. Were incomplete NPCs excluded or just the missing protomers of the incomplete NPCs? Furthermore, did the authors exclude 1252 protomers (80% of their data), or did 1252 protomers remain?

Subtomogram averaging:

"Resolution was measured using the 0.5 criterion"

-> The authors should clarify the rationale for using a non-standard threshold of 0.5 instead of 0.143. In principle, this threshold should only be used when comparing an experimental, noisy map against a noiseless map such as that derived from a PDB model (Rosenthal and Henderson 2003; <https://doi.org/10.1016/j.jmb.2003.07.013>). Were two independent half-maps used for FSC estimation?

Subtomogram averaging:

"After this step NPCs were classified by manually inspecting the sub-volumes into three classes containing CR/IR/NR (27 NPCs), IR/NR (53 NPCs) or IR-only (83 NPCs)."

->The authors should provide examples of how individual NPCs from the tomograms look to warrant manual assignment into these three classes. Why was automated classification of such large structural differences not possible?

Model fitting:

"To fit the Inner Ring, we divided the previously published model (PDB 5IJN) into subcomplexes and used the Chimera "Fit in map" function."

-> It should be clarified whether just a single run of "Fit in Map" from UCSF Chimera was performed, or if multiple random starting positions were attempted and evaluated (<https://www.cgl.ucsf.edu/chimerax/docs/user/commands/fitmap.html>). Please see supplementary material in Mosalaganti et al. 2018 (Figs. S6-69) for a description of how these fits can be validated. Fig. 3b, Extended Fig. 3b,c and Extended Fig. 4 all suggest that the fitting of individual chains may be sub-optimal. Because of the significantly different dimensions of the human NPC presented here compared to the previous model (PDB 5IJN), we recommend that the authors try more powerful methods capable of multi-body fitting into EM maps, such as HADDOCK (<https://www.bonvinlab.org/software/haddock2.4/cryoEM/>) and γ -TEMPy (https://tempy.ismb.lon.ac.uk/multi_component_fitting.html), or more general integrative modelling tools such as IMP (<https://integrativemodeling.org/>) and Assemblin (<https://doi.org/10.1101/2021.04.06.438590>).

Visualization:

"Local resolution analysis was performed as previously reported."

-> Again, this should be described in more detail instead of just pointing to a reference. Also, that reference says that "local resolution was measured with two different techniques". Which of them was used here? Furthermore, was the local resolution information useful for modelling and/or interpretation? If yes, it should be stated. And were the maps filtered by local resolution? Judging from Extended Data Fig. 1c, it doesn't seem to be the case, or the difference is not perceptible.

NPC diameter analysis:

"For diameter measurements, single NPCs were measured using orthoslices. For the DLD1 and HeLa datasets, aligned NPCs from full-NPC averaging were used."

-> Do the authors mean that they measured individual NPC sub-volumes after they were aligned? The authors should clarify this statement because the reader might mistakenly think that the averages were measured.

Data Availability:

-> The authors should deposit their PDB models associated with the EMDB maps. They are largely building upon previously available structures, in particular PDB 5IJN. Future studies should be able to compare and build upon the models derived from the observations presented in this study, for example, the spacing of the inner ring subunits and the reorientation of the outer ring Y-complexes.

MINOR TEXT ISSUES AND TYPOS:

The description of the results seems a bit repetitive/redundant. For example, the dilated inner ring is described in the introduction (lines 38-39) and then twice in the results (lines 64-68, 112-125). These descriptions could be consolidated to improve the flow of the paper.

Line 131: "suggesting they ARE stably anchored in the complex"

Lines 181-183: "To date, the most accurate models are composite structures that combine crystal or EM structures and structure-based models of subcomplexes with cryo-ET data of semi-purified, intact NPCs." -> This statement requires references.

Lines 205-207: "Therefore, taking the NPC out of its native environment might influence the structure more than it if it were a more compact or rigid structure, like, for example, the ribosome." -> We are not sure that the ribosome is a good example of a "compact or rigid structure", as it undergoes a variety of conformational changes that can be resolved by in situ cryo-ET and would be influenced by isolation. The comparison feels unnecessary and can be removed.

Extended Data Figure 4: The Low- and High-threshold descriptions seem to be inverted in panels (d) and (e). That is, more density should be visible at LOW threshold.

Reviewed by Benjamin Engel and Ricardo Righetto

Referee #3 (Remarks to the Author):

Summary of the key results

In this study, the authors analyze nuclear pore complex (NPC) architecture by cryo-electron tomography on cryo-focused ion beam milled human tissue culture cells. In contrast to previous studies using cryo-electron tomography on isolated nuclear envelopes from cells, it can be assumed that the NPCs are better preserved in their native environment. The main findings are that the diameter of the NPC structure is wider than previously observed mainly caused by a different arrangement of the so-called inner ring of the NPC. Second, the components of the cytoplasmic and nuclear ring orient with respect to the pore membrane differently as previously suggested. This also results in an asymmetric membrane curvature at the NPC whereas previously models suggested the symmetric appearance of the curved pore membrane with respect to the nuclear envelope plane. Third, degran-mediated removal of the Y-complex component Nup96, which removes the cytoplasmic or both the nuclear and cytoplasmic ring does largely impact the

inner ring. This suggests that the three-ring structures of NPCs can act largely independently from each other, which could play a role in adapting NPC structures to the cellular needs.

Originality and significance: if not novel, please include reference

Changes in the NPC diameter have been reported for budding and fission yeast but also upon transport of HIV particles through human NPCs (all referenced in the manuscript). Nevertheless, the data presented here, in addition to the unanticipated asymmetric pore membrane arrangement, will be an important step forward towards a refined NPC structure. It might be argued that the previously reported more narrow NPC structure is caused by the sample preparation method.

Data & methodology: validity of the approach, quality of data, quality of presentation

The data are clearly presented. Unfortunately, the main finding on the native NPC structure is derived from NPC where Nup96 is replaced by a Nup96-neonFP-AID fusion protein. It is unclear (and not discussed) how this large add-on would affect the structure of the Y-complexes, and hence of the cytoplasmic and nuclear rings. Given that the large NPC diameter is also seen in mouse fibroblasts in HeLa (torsin-KD cells) this might be a rather safe conclusion. However, the statements on the Y-complex orientations and asymmetric pore membrane appearance should be preferentially deduced from "wild type" DLD-1 cells.

Appropriate use of statistics and treatment of uncertainties

As far as I can judge is all properly handled.

Conclusions: robustness, validity, reliability, Suggested improvements: experiments, data for possible revision

The conclusions are well based on the presented data. However, as indicated above it would be more appropriate to use wild-type DLD-1 cells for the analysis of unperturbed NPCs. If done I think the paper will be of major importance not only for the field of nuclear transport but also as a reference point for cell biology in general.

References: appropriate credit to previous work?

Previous work is properly cited.

Clarity and context: lucidity of abstract/summary, appropriateness of abstract, introduction and conclusions

The manuscript is clearly written and easy to follow. The abstract summarizes the important findings. I am not sure whether the differences seen in NPC structure as compared to previous reports reflect changes in the cellular environment. It might be rather differences in the preparation method. The title might thus need to be adapted.

Author Rebuttals to Initial Comments:

Referee #1 (Remarks to the Author):

In this study, the authors use FIB milling and cryo-ET to examine the structure of the in situ human NPC and the effects on it of depletion of a major outer ring component. The structure of NUP96::Neon-AID labeled human NPCs was mapped by FIB-cryo-ET first, and showed that – by comparison with previous studies on isolated human and other vertebrate NPCs, but in agreement both with previous studies on human NPC and comparisons of yeast isolated versus in situ NPCs, significantly larger inner ring diameters are observed in situ. Moreover, as with Mahamid et al., 2016 (PMID: 26917770), they observe that individual NPCs in situ appear to come in a wide range of inner ring diameters. They also observe an asymmetric curvature of the NE membrane in the nucleocytoplasmic axis, which (also consistent with prior studies in yeast) the authors point out is likely conferred by the NPC ring substructures. While others have studied both in situ and isolated NPCs in vertebrates, explicit comparisons were not made in as much detail as here.

To further analyze the interdependence of the different NPC rings, the authors compare their initial map with the ones obtained using a similar approach upon auxin-induced degradation of the outer ring component Nup96 (derived by autoproteolytic cleavage of a 186 kDa precursor into Nup98 and Nup96), which forms the central stalk of the Y-shaped Nup107-160 complex comprising outer rings. The effects of Nup96 degradation, compared with that of degradation of other outer ring nups (Nup107 and Nup160) and two inner ring Nups (Nup188 and Nup93) were previously examined in an extremely elegant Biorxiv preprint by some of the same authors (Regmi et al., 2020), that comprehensively employed fluorescence microscopy, proteomics and scanning electron microscopy to convincingly demonstrate a surprising degree of structural independence between the nuclear and cytoplasmic outer rings and the inner ring in mammalian cells.

In the present study, the authors show that upon Nup96 depletion, several NPC forms could be detected, basically differing in the absence of the cytoplasmic and nuclear rings while the inner ring remained essentially unaffected. It is not clear though if the observed Nup96 degradation-induced NPC forms are mature NPCs getting disassembled or dead-ends of failed newly assembled NPCs. However they strongly indicate that the C and N outer rings may restrict the inner ring dimensions, intriguingly acting as molecular “rulers” to define the upper limit of the inner ring’s size, as well as modulate NE membrane curvature.

Overall this is a well performed study with a number of intriguing findings, but rather limited in scope, with many of the conclusions – though given in more detail – being ones already covered by others; in essence, this manuscript revisits the in situ FIB milling cryo-ET of human cells first comprehensively surveyed by Wolfgang Baumeister’s group in Mahamid et al., and to some extent by Martin Beck’s lab in Zila et al, 2021 (PMID: 33571428). In combination with the Regmi et al. studies the two manuscripts collectively tell a very compelling story, but (though this is a decision for the editor) this work alone may be too limited in scope for this Journal. There are also a couple of point that the authors could address:

- There are no metrics reported in the manuscript that could help assess the level of confidence of the models generated by the authors for the ORs and IR. The average resolution of the cryo-ET maps is modest (~3.4 nm), and, in principle, not detailed enough to allow an accurate positioning of the different components of the NPC. Known NPC features are apparently missing from the map density, like the Nup155 units connecting the IR and cytoplasmic OR; is this due to their flexibility, or to heterogeneity, or the type of cell line used in the study? The authors specifically mention the papers from Ori et al., 2013 and Raices et al., 2012 where changes in Nup stoichiometry are detected between mammalian cell lines. However, they choose to assume a certain composition and stoichiometry for the strain they used here – but do they know that this cell line indeed has the same stoichiometry and composition they assume? Taking all these considerations together, the comparison of Nup and subcomplex positions between NPC states (in situ vs isolated NEs) remains a bit speculative and potentially ambiguous.

Response:

Regarding the question about metrics and the scoring of fits, please see response to reviewer #2 who raised the same, valid point. We have incorporated this data into Extended Data Fig. 3 in our revised manuscript.

Regarding the positioning of subcomplexes, they are based on previous studies which resulted in comparable resolution. Moreover, the fit (described below and now Extended Data Fig. 3) between the subcomplex densities and our structure provides additional confirmation for the suggested model. That said, we fully agree that at ~3.4 nm resolution modeling capabilities are limited, and throughout the manuscript we have attempted to match our wording with the data we have. We would argue that the density ascribed to Nup155 in other reconstructions is equally speculative, since the resolution of those studies is not significantly different. The reason we hardly see density in the regions connecting the three rings, is probably best explained by conformational flexibility in that region, since the connecting densities between the rings were detected in lower resolved maps (e.g. Maimon et al. 2012). In our data, we can see a weak connecting density in early binned averages (figure below, red arrowheads), but this region averages out to noise in subsequent steps, further suggesting it is flexible.

Finally, regarding the stoichiometry of individual Nups our modeling approach centers on the most conserved scaffold elements of the NPC, i.e. IR, NR, and CR. For those,

the stoichiometry between widely used mammalian cell lines is the same, as reported by Ori et al. We therefore believe that we used valid assumptions regarding those Nups, consistent with our experimental data.

- It has been shown that in baker's yeast the ORs are only connected through a few flexible linkers to the IR (Kim et al., 2018; Allegretti et al., 2020), and it was suggested that such an arrangement provides flexibility and resilience to the NPC. It would be interesting to discuss and compare those observations with the observations made by the authors in the human in situ NPC. Previous vertebrate NPC maps suggested a much more solid connection between rings.

Response:

We agree with the interesting point made by the reviewer. We have expanded the relevant text passage in the revised manuscript discussion.

Referee #2 (Remarks to the Author):

This manuscript by Schuller et al. uses focused ion beam (FIB) milling followed by cryo-electron tomography (cryo-ET) and subtomogram averaging to resolve a native structure of the human nuclear pore complex (NPC) within DLD-1 colon cancer cells. Compared to previous cryo-ET structures from isolated nuclei, the authors observe that the NPC's inner ring is significantly dilated, and gaps are present between the inner ring's asymmetric subunits (also called "spokes"). Overall, the NPC has a shorter structure along its nucleocytoplasmic axis, with the outer ring Y-complexes angled toward the central channel. The authors also observe asymmetry in the nuclear envelope, which they infer may be caused by asymmetric forces from the two outer rings. Finally, they used cryo-ET to generate low-resolution averages of NPCs in various states of disassembly following Nup96-depletion, which was previously shown to disassociate the outer rings from the NPC. From this depletion experiment, they propose that the nuclear ring is more firmly attached to the NPC than the cytoplasmic ring, and that the outer rings limit the maximum dilation of the inner ring.

We have a mixed opinion of this manuscript. On the positive side, this appears to be the best-resolved in situ human NPC structure to date. While the estimated resolution of the subtomogram averaging (34 Å) is not a dramatic improvement on a previously published in situ human NPC (37 Å) (Zila et al., 2021; Cell), the densities in the map certainly appear to be more defined. As the authors acknowledge (lines 248-249), the moderate resolution of their structure is insufficient to model molecular structures into significant portions of their NPC density map (only the Y-complexes are modeled in the cytoplasmic and nuclear rings). Thus, future improvements will be required to produce a more "definitive" human NPC structure. However, the NPC is a challenging target due to its size and flexibility, and the success of this study should not be undervalued-- the structure presented in this manuscript is the current benchmark for native NPC architecture in human cells.

NOVELTY:

The biggest issue with this manuscript has to do with how it is presented. Throughout the abstract, introduction, and results, the authors assert the novelty of the paper's main finding: the NPC's inner ring is more dilated inside the cell than in isolated nuclei. Only in the discussion section do the authors begin to reveal that this has been well documented before (lines 187-190: "The ability to modulate its diameter may be a general property of NPCs as we observed this phenomenon in mouse embryonic fibroblasts, and it has also recently been reported in *S. cerevisiae* [34] and in various growth conditions for *S. pombe* [39]"; lines 212-214: "A recent study of HIV infected T cells is consistent with such IR flexibility and revealed a similarly widened NPC channel, large enough to accommodate the intact HIV-1 capsid [42]").

To the best of our knowledge, there are FOUR previous studies that have resolved in situ structures of NPCs inside cells using a combination of FIB milling and cryo-ET. These studies all describe the asymmetry of the NPC and dilation of the highly-

conserved inner ring. Their results should be presented in the introduction so the reader can understand the current state of knowledge in the field and how Schuller et al. adds to it. We list these studies below:

1) Mosalaganti et al., 2018; Nature Comm; <https://doi.org/10.1038/s41467-018-04739-y>
This study presents the native structure of the *C. reinhardtii* NPC (resolution: 30 Å) and provides an in-depth comparison to the isolated human NPC structure (with figures that are quite similar to Fig. 3 here in Schuller et al.). The dilation of the inner ring and resulting “peripheral channel” spaces between the spokes are shown in detail. The reduced NPC height vs. the isolated human NPC is also clearly documented. The authors propose that this is a species-specific difference. However, as Schuller et al. observe reduced height of the human NPC in situ, this algal NPC result could be reconsidered in the discussion.

Key descriptions:

“The inner ring of the CrNPC map is dilated in comparison to the HsNPC map, with substantial spacing between its rotationally symmetric spokes, thereby forming relatively large peripheral channels that have been proposed to accommodate the import of inner nuclear membrane proteins.”

“Are these species-specific differences in the inner ring, or could they be related to the NPC’s functional state? ...We speculate that not only the FG-rich regions, but also the scaffold of the NPC may be much more dynamic than anticipated.”

2) Allegretti et al., 2020; Nature; <https://doi.org/10.1038/s41586-020-2670-5>
This study presents a well-resolved native structure of the *C. cerevisiae* NPC (resolution: 25 Å). The dilated inner ring and resulting spaces between the spokes are very clearly shown.

Key description:

“The entire IR has to be dilated by about 20 nm in diameter, thereby spatially separating the eight individual spokes. This analysis underlines the plasticity of the NPC within cells, which might be physiologically relevant for the transport of large cargos and inner nuclear membrane proteins.”

3) Zimmerli et al., 2020; bioRxiv; <https://doi.org/10.1101/2020.07.30.228585>
This study presents a well-resolved native structure of the *C. pombe* NPC (resolution: 23 Å). We consider this to be the current definitive paper on inner ring dilation, as it demonstrates how the inner ring dilates and contracts in response to the cell’s energy state. The spaces between the inner ring spokes are also very clear in the dilated NPC structure.

Key descriptions:

“The central channel constriction of the IR is more elaborate and mediated by a lateral displacement of the 8 spokes that move as independent entities to constrict or dilate the IR. In the dilated state, around 3-4 nm wide gaps are formed in between the neighboring spokes, while in the constricted state the spokes form extensive contacts, equivalent to those in the previously published structures of the human NPC in isolated nuclear envelopes”.

“Our dynamic structural model suggests large scale conformational changes that occur by movements of the spokes with respect to each other but largely preserve the arrangement of individual subcomplexes. Previous structural models obtained from isolated nuclear envelopes thereby represent the most constricted NPC state”.

4) Zila et al., 2021; Cell; <https://doi.org/10.1016/j.cell.2021.01.025>

This study presents the native structure of the human NPC (resolution: 37 Å) and describes in detail how the inner ring is dilated in both HIV-infected and non-infected cells. This confirms the conservation of inner ring dilation from algae and yeast to humans. The spaces between the inner ring spokes are visible in the subtomogram average (Fig. S7B). The reduced height of the NPC in situ is also apparent. The membrane curvature is asymmetric across the NPC (Fig. 6C), but the angles do look different than what is shown here in Schuller et al., which could be a point of discussion.

Key description:

“In line with other studies conducted in intact human cells (Beck and Baumeister, 2016; Mahamid et al., 2016), the NPCs appeared dilated in comparison to isolated nuclear envelopes and displayed an average diameter of ~64 nm. ...We found that the NPCs dilated to similar diameters in non-infected T cells. Taken together, our findings indicate that the NPC structure observed under the relevant conditions, namely in infected and non-infected T cells in situ, is representative of the transporting state, whereas the constricted state observed in isolated nuclear envelopes (von Appen et al., 2015) may be more relevant to stress conditions (Zimmerli et al., 2020).”

In light of the above studies, the following statements in Schuller et al. overstate their novelty and should be reworded to put their observations into context with the NPC field:

Abstract: “Our findings highlight the inherent flexibility of the NPC and strongly suggest that the cellular environment has a hitherto underappreciated influence on NPC dimensions and architecture.”

-> How can the in situ effect on NPC architecture possibly be “hitherto unappreciated”?

Lines 38-39: “Surprisingly, the organization of the IR increases the central pore

diameter by more than 30%”

-> How is this a surprise? It's to be expected based on previous studies.

Lines 43-44: “Our study shows that the cellular environment significantly influences the diameter of the central channel of the NPC”

-> We would say that it corroborates this previously-reported effect of the cellular environment.

Lines 183-185: “Our study reveals a substantial NPC plasticity and shows that purification of NPCs and the NE influences the NPC structure.”

-> Rather than “reveals”, we would say that this study strongly confirms the previous hypothesis (from algae, yeast, and human cells) that purification of NPCs and the NE influences the NPC structure.

Lines 211-216: “Furthermore, we observe that the subunits in the IR contain gaps between adjacent subunits, while previous models suggested a compact, interwoven architecture. ...This flexibility could be important for protein trafficking to the INM which requires passage through the NPC while membrane anchorage is retained.”

-> As we outline above, all four previous studies have observed “peripheral channel” gaps between the spokes of the dilated inner ring, and thus should be correctly cited. The idea that these gaps could permit passage of membrane proteins through the NPC has also been proposed:

...proposed in Mosalaganti et al. 2018: “The inner ring of the CrNPC map is dilated in comparison to the HsNPC map, with substantial spacing between its rotationally symmetric spokes, thereby forming relatively large peripheral channels that have been proposed to accommodate the import of inner nuclear membrane proteins.”

...proposed in Allegretti et al., 2020: “This analysis underlines the plasticity of the NPC within cells, which might be physiologically relevant for the transport of large cargos and inner nuclear membrane proteins.”

...proposed in Zimmerli et al., 2020: “Peripheral channels are thought to be important for the nuclear import of inner nuclear membrane proteins. Here we observed around 3-4 nm wide lateral gaps between the individual spokes of actively transporting NPCs. Notably, our data processing workflow yields an average of conformation under the respective conditions and individual spokes are even more dynamic. Therefore, it is plausible that the opening and closing of peripheral channels may regulate the translocation of inner nuclear membrane proteins.”

Lines 216-222: “A wider central channel does not only have implications for the maximal size of soluble and membrane-bound cargo that can pass through the NPC, but it also changes the density of the FG-repeats that fill the transport channel. The potential of the FG density to influence transport has been documented in model systems that focused on this aspect of NPC biology. Our NPC model indicates that the FG-density in the central channel is lower than previously thought, or that flexibility, particularly within the IR, may be a means to modulate the properties and selectivity of the FG-barrier.”

-> This hypothesis of inner ring dilation changing FG concentration has also been proposed before, and should be cited as such:

...proposed in Mosalaganti et al. 2018: "The local FG-Nup concentration within the central channel might change during inner ring dilation."

...proposed in Zimmerli et al., 2020: "The estimated volume of the central channel in the most dilated state was almost twice as large (~152'000 nm³) as compared to the most constricted state (~86'000 nm³), which likely translates to ~2-fold change in concentration of the FG-repeats contained therein." "It therefore is plausible that a constricted central channel volume leads to an increased local FG-domain concentration which in turn limits the passive diffusion of molecules of a constant size."

Lines 227-228: "Our data might indicate that the CR/NR act as molecular "rulers" to define the upper limit of the IR dimensions."

-> Again, this hypothesis has been proposed before and should be cited:

...proposed in Mosalaganti et al. 2018: "The CrNPC inner ring has the same diameter as the outer rings, which are horizontally stacked upon it. In this conformation, the head-to-tail connection of the outer ring Y-complexes might be important for restricting the maximal dilation of the pore."

In summary, the main findings in this paper build on previous work and must be presented in that context. The good news is that this can be relatively easily addressed by reworking the text of the manuscript. There is the perfect opportunity to discuss how the in situ architecture and dilation of the inner ring are conserved between species (algae, yeast, humans), while the outer rings vary widely. As for the question of "novelty", while it is true that many of the main findings in this paper have been reported in some fashion before, we do not believe this should prohibit publication. The improved structure of the human NPC is an important advance and adds a new layer to our understanding of NPC architecture across the tree of life.

Response:

We appreciate the reviewer's thorough comparison with previously published work in the field. It was not our intention to take away from those studies, and we agree that the way they occur in the text did not give these other studies due credit. Also, omitting the Mosalaganti et al. study in our manuscript was a regrettable oversight that we have corrected.

Nevertheless, we want to point out a few things: First and foremost, it is important to separate the discussion about human NPCs from those studies that involve other organisms. Three of the four studies mentioned involved *S. cerevisiae*, *S. pombe* and *C. reinhardtii*, respectively. These are single-cell organisms, each evolutionarily more than a billion years separated from humans, with different Nup compositions and therefore, unsurprisingly, different dimensions, including diameter. The thrust of all these three

papers, appropriately, is the respective difference to the human NPC, which is a different message than our paper has. The Zila et al. study, which focused on HIV-1 capsids and their transport through the NPC in T cells, does not provide a detailed analysis of the different NPC subcomplexes and membranes, the effect of purifications, nor the asymmetry in the pore organization. Most importantly, the structural flexibility of the human NPC was not discussed as a result of changes in the cytoskeleton and cellular microenvironment as we propose. The proposed hypothesis in Zila et al. is that partially purified NPC are in a 'stressed' condition, an idea that was further detailed in the *S. pombe* work by Zimmerli et al. But, respectfully, we find it to be quite a stretch to equate energy depletion as observed in *S. pombe* and its effect on the NPC diameter, with the observations we make in mammalian cells.

We agree that some of the concepts we propose were mentioned in those previous papers, but they by-and-large could have been explained by other phenomena (species-specific differences, etc.) as well. However, we revised the text and adjusted the referencing in a way that those previous studies are now clearly mentioned.

Specifically, we have:

1. Introduced the Allegretti, Mosalaganti, Zimmerli, and Zila studies in the introduction, better placing our study in the context of the current state of knowledge (lines 37-42).
2. Added an additional statement in the discussion to better reference and give credit to the Zila et al. and Zimmerli et al. studies that also observe a larger NPC diameter than anticipated, or in the context of different cellular states (lines 210-211).
3. Added a proper citation for the Mosalaganti et al. and Allegretti et al. works that suggested the IR peripheral channels may be important for INM transport (lines 220-221).
4. As above, added a citation to the Mosalaganti et al. and Zimmerli et al. work for proposing impacts on the FG-density based on IR changes (lines 227-228).
5. Cited Mosalaganti for proposing the CR/NR may restrict the maximal pore dimensions (lines 234-235).
6. Added a note in our discussion regarding the potential difference in membrane curvature that appears to be observed in HIV-infected cells from the Zila et al. manuscript (lines 259-264). We refrained from making any speculations about this potential difference, however, because the deposited map comes from HIV-infected cells rather than uninfected cells.

TECHNICAL ISSUES:

In addition, there are several technical issues found in the Methods section that should be addressed:

Tilt series acquisition and processing:

"CTF was determined and corrected as described previously"

-> Such a crucial step should be described in the paper instead of pointing to an external reference. Was it done by phase flipping? CTF multiplication? Wiener filtering? Was it applied in 2D (defocus gradient on images of tilt series) or was it a 3D correction (e.g., NovaCTF)? Which software was used? (cite it!). These details should be given.

Response:

We have added a more detailed summary of CTF determination and correction inside our Methods section.

Subtomogram averaging:

“At this stage incomplete NPC sub-volumes were excluded based on manual inspection (1252 protomers).”

-> It is not clear here what was excluded. Were incomplete NPCs excluded or just the missing protomers of the incomplete NPCs? Furthermore, did the authors exclude 1252 protomers (80% of their data), or did 1252 protomers remain?

Response:

We thank the reviewer for catching this and have fixed the text accordingly. At this stage, single protomers were manually inspected and missing protomers from incomplete NPCs, misaligned protomers, and protomers with low signal-to-noise were excluded. After this step, 1252 protomers remained for subsequent steps.

Subtomogram averaging:

“Resolution was measured using the 0.5 criterion”

-> The authors should clarify the rationale for using a non-standard threshold of 0.5 instead of 0.143. In principle, this threshold should only be used when comparing an experimental, noisy map against a noiseless map such as that derived from a PDB model (Rosenthal and Henderson 2003; <https://doi.org/10.1016/j.jmb.2003.07.013>). Were two independent half-maps used for FSC estimation?

Response:

In the revised manuscript we also indicated the 0.143 threshold but we think that the more conservative estimate at FSC 0.5 better reflects the actual resolution. Since we performed sub-tomogram averaging by merging the two half-sets after each iteration as the template for the next iteration, the FSC 0.5 criterion is presumably a better estimation (and a more modest one) of the true resolution.

Subtomogram averaging:

“After this step NPCs were classified by manually inspecting the sub-volumes into three classes containing CR/IR/NR (27 NPCs), IR/NR (53 NPCs) or IR-only (83 NPCs).”

->The authors should provide examples of how individual NPCs from the tomograms look to warrant manual assignment into these three classes. Why was automated classification of such large structural differences not possible?

Response:

We have included examples of single NPCs in Extended Data Fig. 6c. Since the differences are stark and can be detected at the level of the individual raw tomograms, and the dataset is relatively small (<200 NPCs), we did not attempt any automated classification approaches. We also provided the averages of the classes after our manual classification and rough manual alignment in Extended Data Fig. 6d. The class-averages clearly show the 3-, 2-, or 1- ring architectures and confirm that our subjective, manual classification represent the individual particles well.

Model fitting:

“To fit the Inner Ring, we divided the previously published model (PDB 5IJN) into subcomplexes and used the Chimera “Fit in map” function.”

-> It should be clarified whether just a single run of “Fit in Map” from UCSF Chimera was performed, or if multiple random starting positions were attempted and evaluated (<https://www.cgl.ucsf.edu/chimerax/docs/user/commands/fitmap.html>). Please see supplementary material in Mosalaganti et al. 2018 (Figs. S6-69) for a description of how these fits can be validated. Fig. 3b, Extended Fig. 3b,c and Extended Fig. 4 all suggest that the fitting of individual chains may be sub-optimal. Because of the significantly different dimensions of the human NPC presented here compared to the previous model (PDB 5IJN), we recommend that the authors try more powerful methods capable of multi-body fitting into EM maps, such as HADDOCK (<https://www.bonvinlab.org/software/haddock2.4/cryoEM/>) and γ -TEMPy (https://tempy.ismb.lon.ac.uk/multi_component_fitting.html), or more general integrative modelling tools such as IMP (<https://integrativemodeling.org/>) and Assemblin (<https://doi.org/10.1101/2021.04.06.438590>).

Response:

We thank reviewers #2 and #1 for this comment and provide the following analysis of our models using the Mosalaganti et al. 2018 reference as a guide. We have included this analysis in the Methods section of our paper and the results of this analysis as Extended Data Figure 3 and further details in our Methods. The results are summarized here as well.

To create our models for the Y complexes in the CR/NR and the IR complex, we performed unbiased global fitting using structural models derived from previously reported human NPC structures (von Appen et al. 2015, Kosinski et al. 2016). The fitting was performed independently for the Cytoplasmic and Nuclear Rings using a 3-protomer segmented model since our single protomers do not necessarily contain a complete protomer of the 8-fold symmetric NPC (i.e. a single protomer may contain $\frac{1}{2}$ of the Y complex from adjacent protomers). For the Inner Ring, a single protomer was used since it contained the entire complex. All fitting runs were performed using Chimera and 1,000,000 random initial placements and local cross-correlation (Chimera’s correlation about the mean, CAM), or a combination of local cross-correlation and overlap (CAM+OVR) was used when local cross-correlation did not provide statistically significant fits (also noted in Mosalaganti et al. 2018). Our IR

assembly scored 0.873, and our Y complex models match the locations of the previous model which had scores of 0.728 and 0.756.

For the IR complex (Extended Data Fig 3d-e), we identified a significant fit in the expected position as described in the most recent human structure (PDB 5IJN, Kosinski et al. 2016), though it was obvious that some of the nucleoporins in this model could be shifted to better fit our density map upon closer inspection. We therefore employed a similar approach to Mosalaganti et al. study and optimized the fits by local re-fitting of individual subunits or domains. After this local re-fitting, we decided to re-run the unbiased global fitting approach as described, and again find a significant fit in the same location where the previous model was docked (with a slightly higher CAM score).

For the Y complex (Extended Data Fig 3a-c) we identified a significant fit in the expected position as described in the most recent human NPC structure CR and NR (PDB 5A9Q, von Appen et al. 2015) using a double Y complex as the reference molecule. Since the 5A9Q model contains gaps in the heterotetrameric core element of the Y complex called the “hub” (Nup160, 85, 96, and Sec13) we decided to replace the hub of the 5A9Q model with the hub of a new composite model we created using published x-ray crystallography structures and threaded models. The hub of this updated model was then used to rigid-body dock into the CR/NR of our map at the same position that the 5A9Q hub occupied, and the Nup107-133 subcomplex from 5A9Q was kept as is, creating a new complete Y complex assembly. Additional local re-fittings were performed as noted previously. After the final model creation for both the CR and NR, we similarly ran another round of unbiased global fitting using a single protomer of Y complexes (a double Y complex) and identified a similar location where the Y-complex of 5A9Q docked. A final local optimization run maximized the density overlap.

Comparing the two models after this final optimization step, it is readily apparent how similar our model is compared with the previous 5A9Q model, other than the regions of the hub that we have now included. We also tried to fit smaller portions of the Y complex (single Y complexes or the hub only), but in all these cases the data did not reveal any statistical significance though the top fits docked exactly into the positions we would expect based on the previous model and our current model.

We are therefore confident in our models, especially since they overlap well with the previous published models, except for some deviations by Nups that can account for the overall architectural differences we observe between our native structure and the semi-purified structure. However, we strongly urge that the final fits are not interpreted at atomic resolution. Instead, our fitting simply confirms the assignment of the densities toward understanding how the subcomplexes are positioned.

Visualization:

“Local resolution analysis was performed as previously reported.”

-> Again, this should be described in more detail instead of just pointing to a reference.

Also, that reference says that “local resolution was measured with two different techniques”. Which of them was used here? Furthermore, was the local resolution information useful for modelling and/or interpretation? If yes, it should be stated. And were the maps filtered by local resolution? Judging from Extended Data Fig. 1c, it doesn't seem to be the case, or the difference is not perceptible.

Response:

We have included a summarized description of this analysis in our Methods section. We measured the local resolution of our map to better assess the resolution provided by FSC analysis but did not use this information for any modeling or structural conclusions. We did not filter the maps by resolution, but simply believe every map should be accompanied by a local resolution analysis.

NPC diameter analysis:

“For diameter measurements, single NPCs were measured using orthoslices. For the DLD1 and HeLa datasets, aligned NPCs from full-NPC averaging were used.”

-> Do the authors mean that they measured individual NPC sub-volumes after they were aligned? The authors should clarify this statement because the reader might mistakenly think that the averages were measured.

Response:

We thank the reviewers for this comment and have adjusted the text accordingly. Yes, we measured the diameter using individual NPC volumes, at the level of the pore membrane.

Data Availability:

-> The authors should deposit their PDB models associated with the EMDB maps. They are largely building upon previously available structures, in particular PDB 5IJN. Future studies should be able to compare and build upon the models derived from the observations presented in this study, for example, the spacing of the inner ring subunits and the reorientation of the outer ring Y-complexes.

Response:

We were hesitant to deposit PDB files of our models since they are not a new “ground-truth” for the IR or Y complex subassembly structures, and we realize that the resolution of our structure requires that the final fits not be interpreted at atomic resolution. However, we have decided to make the models available on Zenodo, an open-source data deposition platform at the following DOI which will be made public upon publication: [10.5281/zenodo.5039644](https://doi.org/10.5281/zenodo.5039644).

MINOR TEXT ISSUES AND TYPOS:

The description of the results seems a bit repetitive/redundant. For example, the dilated inner ring is described in the introduction (lines 38-39) and then twice in the results (lines 64-68, 112-125). These descriptions could be consolidated to improve the flow of

the paper.

Line 131: "suggesting they ARE stably anchored in the complex"

Lines 181-183: "To date, the most accurate models are composite structures that combine crystal or EM structures and structure-based models of subcomplexes with cryo-ET data of semi-purified, intact NPCs." -> This statement requires references.

Lines 205-207: "Therefore, taking the NPC out of its native environment might influence the structure more than it if it were a more compact or rigid structure, like, for example, the ribosome." -> We are not sure that the ribosome is a good example of a "compact or rigid structure", as it undergoes a variety of conformational changes that can be resolved by in situ cryo-ET and would be influenced by isolation. The comparison feels unnecessary and can be removed.

Extended Data Figure 4: The Low- and High-threshold descriptions seem to be inverted in panels (d) and (e). That is, more density should be visible at LOW threshold.

Reviewed by Benjamin Engel and Ricardo Righetto

Response:

All minor text issues were addressed as suggested, and Extended Data Fig. 4 (now Extended Data Fig. 5) corrected.

Referee #3 (Remarks to the Author):

Summary of the key results

In this study, the authors analyze nuclear pore complex (NPC) architecture by cryo-electron tomography on cryo-focused ion beam milled human tissue culture cells. In contrast to previous studies using cryo-electron tomography on isolated nuclear envelopes from cells, it can be assumed that the NPCs are better preserved in their native environment. The main findings are that the diameter of the NPC structure is wider than previously observed mainly caused by a different arrangement of the so-called inner ring of the NPC. Second, the components of the cytoplasmic and nuclear ring orient with respect to the pore membrane differently as previously suggested. This also results in an asymmetric membrane curvature at the NPC whereas previously models suggested the symmetric appearance of the curve pore membrane with respect to the nuclear envelope plane. Third, degron-mediated removal of the Y-complex component Nup96, which removes the cytoplasmic or both the nuclear and cytoplasmic ring does largely impact the inner ring. This suggests that the three-ring structures of NPCs can act largely independently from each other, which could play a role in adapting NPC structures to the cellular needs.

Originality and significance: if not novel, please include reference

Changes in the NPC diameter have been reported for budding and fission yeast but also upon transport of HIV particles through human NPCs (all referenced in the manuscript). Nevertheless, the data presented here, in addition to the unanticipated asymmetric pore membrane arrangement, will be an important step forward towards a refined NPC structure. It might be argued that the previously reported more narrow NPC structure is caused by the sample preparation method.

Data & methodology: validity of the approach, quality of data, quality of presentation

The data are clearly presented. Unfortunately, the main finding on the native NPC structure is derived from NPC where Nup96 is replaced by a Nup96-neonFP-AID fusion protein. It is unclear (and not discussed) how this large add-on would affect the structure of the Y-complexes, and hence of the cytoplasmic and nuclear rings. Given that the large NPC diameter is also seen in mouse fibroblasts in HeLa (torsin-KD cells) this might be a rather safe conclusion. However, the statements on the Y-complex orientations and asymmetric pore membrane appearance should be preferentially deduced from "wild type" DLD-1 cells.

Response:

The use of GFP or homologous ~27 kDa proteins as fusions is common practice in the nuclear pore field for live-cell imaging. There are numerous studies that show that these additions by and large have no effect on the NPC architecture, across the phylogenetic spectrum from yeast to human. Perhaps the most relevant to this study is the Thevathasan et al. study in Nature Methods 2019 from the Ellenberg and Ries labs at EMBL in Heidelberg, Germany (doi.org/10.1038/s41592-019-0574-9) or the Schlichthaerle et al. study in 2019 from the Jungmann lab at MPI-Biochemistry in Martinsried, Germany (<https://doi.org/10.1002/anie.201905685>). Here, Nup96 was

labeled with four different fluorescent tags and used as reference standard for superresolution microscopy in U2OS. Another recent example is the Otsuka et al. study on different NPC assembly modes, as observed in genome-edited HeLa cells (doi.org/10.1101/2021.05.17.444137). In that study, ten different Nups were genetically FP-tagged without noticeable effects to the NPC (Fig.1 of that study).

In our case, we know exactly where Nup96 is positioned in the Y-complex, because of the available crystal structures and reliable homology models deduced from them. A C-terminal addition of ~ 50kD (mNeon-FP ~27 kD, 3x-miniAID, ~23 kD), flexibly attached with an 18 amino acid Ser-Gly-rich linker, is unlikely to interfere with the binding of Nup107 or Sec13, the neighbors to Nup96 in the Y complex. It should also not interfere with the interaction of neighboring Y complex in the 16-membered CR and NR rings, because of the very open, net-like structure of these assemblies. To illustrate this, we include a figure below.

The mNeon-3X-miniAID moiety is flexibly attached using an 18-amino acid Ser-Gly-rich linker to the C-terminus of Nup96 (cyan). In the Y complex model, (created from x-ray crystallographic data) the C-terminal 15 amino acids are unresolved, as they are unstructured. Together, this creates a >10 nm flexible linker to attach this moiety on Nup96.

As the reviewer also points out, the wider NPC IR diameter is also observed in MEFs and in HeLa (TorsinA/B/3A knockout) cells, which, we agree, should serve well as a support of our conclusions. In fact, we would argue that the conservation across different cell types is an even stronger argument for the larger IR diameter not being influenced by the neonFP-AID fusion.

Furthermore, we show in a figure below how the Y complexes in the CR/NR of the DLD-1 and HeLa structures (pictured is a single protomer of the 8-fold symmetric CR or NR) we determined are nearly identical, further suggesting no impact from this fusion protein.

Given the significant effort in time and money it would take to repeat the reconstruction with wild-type DLD-1 cells, we hope the reviewer can agree that the controls as presented are sufficient to back our general conclusions.

Appropriate use of statistics and treatment of uncertainties
As far as I can judge is all properly handled.

Conclusions: robustness, validity, reliability, Suggested improvements: experiments, data for possible revision

The conclusions are well based on the presented data. However, as indicated above it would be more appropriate to use wild-type DLD-1 cells for the analysis of unperturbed NPCs. If done I think the paper will be of major importance not only for the field of nuclear transport but also as a reference point for cell biology in general.

Response:

We have addressed this concern above.

References: appropriate credit to previous work?
Previous work is properly cited.

Clarity and context: lucidity of abstract/summary, appropriateness of abstract, introduction and conclusions

The manuscript is clearly written and easy to follow. The abstract summarizes the important findings. I am not sure whether the differences seen in NPC structure as compared to previous reports reflect changes in the cellular environment. It might be rather differences in the preparation method. The title might thus need to be adapted.

Response:

We agree with the reviewer that the difference in preparation method may be responsible for the significant differences between our structure and the previously reported structure. We decided to use the term “cellular environment” to be all-encompassing since we cannot pinpoint exactly what is retained using cryo-FIB that impacts the NPC architecture, and is lost during the nuclear purification procedure. We suggest in the discussion that the nucleo- or cyto-skeletal networks, or chromatin itself could impact the structure, but we felt that using the term “cellular environment” would cover all these potential options.

Reviewer Reports on the First Revision:

Referee #1 (Remarks to the Author):

The reviewers have fully addressed my technical queries in the previous review, concerning metrics, the scoring of fits, the positioning of subcomplexes, and the assumptions of stoichiometry – their answers and text amendments are all very reasonable.

I also re-iterate our own comments agreeing with Referee #2 about being continuing to be careful in the next revision of the manuscript about over-stating the novelty of the findings, though appreciate the difficulty in ensuring everyone is cited. However, there are a couple that should be added: (i) Regarding the dilation of the inner ring - as well as the references from Referee #2, Mahamid et al., 2016 should also be cited in this context as they were among the first to demonstrate this (even though a bit buried) and discuss: "Deviations from previous structures were evident for the pore diameter in the spoke ring (Fig. S7C)... This confirmed previous observations of NPC plasticity and illustrates the problem of analyzing large flexible structures by averaging methods". Feldherr also published a couple of papers about the ability of NPCs to alter their functional diameter (e.g. Feldherr et al., 1998), which may also be of interest in this regard. (ii) Regarding potential membrane transport through gaps between the spokes – as well as the references from Referee #2, there is also Kim et al., 2018 that could also be cited in this context as they also saw these channels, stating: "This juxtaposition of arches and transient openings may delineate conduits for nucleocytoplasmic transport of transmembrane proteins, potentially resolving the issue of how membrane proteins transit the NPC".

Referee #2 (Remarks to the Author):

The authors did an admirable job of addressing the major points raised in our review, improving their modeling and validation, as well as the discussion of previous in situ studies. The NPC model appears to be significantly improved and fits better into the experimental data, as shown in Extended Data Figure 3. Extended Data Figure 6 also now illustrates the manual classification of NPCs with missing nuclear and cytoplasmic rings. Although we believe that this study is nearly ready for publication, a few easily addressed, yet important, issues remain:

DISCUSSION OF PREVIOUS STUDIES:

Introduction, Lines 37-43: "NPCs from cryo-FIB milled single-cell organisms have been studied recently and show significant differences to human cells 28-30. HIV-1 infected human T cells also show differences 31. Here, we study the architecture of the human NPC from cryo-FIB milled DLD-1 cells. Our data establishes concepts that may have been previously interpreted as largely species-specific differences in the context of human cells. The organization of the IR increases the central pore diameter by more than 30%, while the two other ring moieties reorient differently with respect to the nuclear membrane."

While we appreciate that the preceding studies are now mentioned in the introduction, two key pieces of information are missing that set the stage for the current work:

1) The preceding in situ NPC structures don't just show "differences". It should be specifically mentioned in the introduction that all these studies observed a dilated inner ring diameter compared to structures from isolated NPCs. As far as the interpretations in the previous studies, both species-specific and physiological differences have been proposed for the dilated inner ring. The current study proves that it is the latter.

2) The study by Zila et al. (reference 31) observed equally dilated inner rings in both infected and non-infected human cells. Thus, the dilated inner ring has already been shown to be a feature of human cells, not a specific consequence of HIV infection. This is quantified in Fig. 6D of Zila et al. and described as follows: "To address if HIV-1 infection may promote NPC dilation, we collected cryo-electron tomograms of non-infected SupT1-R5 cells and measured the diameter of 39 NPCs. We found that the NPCs dilated to similar diameters in non-infected T cells (Figure 6D). Taken together, our findings indicate that the NPC structure observed under the relevant conditions, namely in infected and non-infected T cells in situ, is representative of the transporting state, whereas the constricted state observed in isolated nuclear envelopes (von Appen et al., 2015) may be more relevant to stress conditions (Zimmerli et al., 2020)."

While the authors are of course free to pick their own wording incorporating the above information, we suggest the following:

"NPCs from cryo-FIB milled yeast and algae cells have been studied recently and show significant differences to previous human NPC structures, including dilated IRs 28-30. Dilated IRs were also observed in cryo-FIB milled human T cells, both when infected with HIV and without infection 31. Here, we study the architecture of the human NPC from cryo-FIB milled DLD-1 cells. Our data establishes concepts in human NPC architecture that may have previously been interpreted as species-specific differences. The dilated organization of the IR increases the central pore diameter by more than 30% compared to previous structures of isolated human NPCs, while the two other ring moieties reorient differently with respect to the nuclear membrane."

DATA AVAILABILITY:

The PDB file containing the atomic model coordinates generated by this study must be deposited at the PDB instead of Zenodo, following the Nature Editorial Policies as stated in:

<https://www.nature.com/nature/editorial-policies/reporting-standards#availability-of-data>

<https://www.nature.com/sdata/policies/repositories#molec>

"Mandatory deposition of data is required for certain data types; Macromolecular structure: Worldwide Protein Data Bank (wwPDB)".

It is evident that a model generated from a map at this resolution level ($\sim 34 \text{ \AA}$) should not be interpreted at atomic resolution. Nevertheless, the relative subunit positions and orientations are of major interest to the scientific community (in our original review, we called it "the current benchmark for native NPC architecture in human cells"). If the authors are concerned that their coordinates may be misinterpreted, they can for example trim the side chains from the model, leaving just the structure backbone, which is understandable at this resolution.

TYPOS:

Introduction, Line 40:

Our data establishes concepts that may have previously BEEN interpreted

Results, Line 172-174:

Although the resolution did not allow fitting the IR model into our map, we placed the model as an estimate for the IR mass into these maps AND OBSERVED THAT the spacing between the IR subcomplexes increases as fewer ring structures are present (Fig. 4c).

Methods, visualization:

"chimeras" should be "Chimera's"

A FINAL NOTE:

Finally, this does not affect our assessment of this paper, but we wanted to respond to the authors' claim that: "it is important to separate the discussion about human NPCs from those studies that involve other organisms. Three of the four studies mentioned involved *S. cerevisiae*, *S. pombe* and *C. reinhardtii*, respectively. These are single-cell organisms, each evolutionarily more than a billion years separated from humans."

While we appreciate the argument that there is great evolutionary distance between the different organisms that have been studied so far, there is nothing special about humans that make them above comparison to other organisms such as algae and yeast, especially for a molecular complex like the NPC, which is a shared defining feature of all eukaryotes. Multicellularity is the wrong metric—it is not indicative of evolutionary distance, and indeed it has arisen numerous times. In terms of genetic drift, algae and yeast are both more closely related to humans than they are to each other (please see the attached "phylogenies.pdf", which plots genetic distance across the tree of life). Unlike the more divergent outer rings of the NPC, the NPC inner ring is highly conserved between algae, yeast, and humans. Thus, the fact that both yeast and algae show a dilated inner ring in situ has accurately foreshadowed that the same would be true in humans. This conclusion is not intended to take anything away from the current study, but this is the clear evolutionary context that these new findings fit into.

Reviewed by Benjamin Engel and Ricardo Righetto

Referee #3 (Remarks to the Author):

The authors have addressed all points raised by the reviewers. The manuscript now clearly reports the state of current knowledge taking into account published work and describes adequately the knowledge gain of this study.

Author Rebuttals to First Revision:

Referees' comments:

Referee #1 (Remarks to the Author):

The reviewers have fully addressed my technical queries in the previous review, concerning metrics, the scoring of fits, the positioning of subcomplexes, and the assumptions of stoichiometry – their answers and text amendments are all very reasonable.

I also re-iterate our own comments agreeing with Referee #2 about being continuing to be careful in the next revision of the manuscript about over-stating the novelty of the findings, though appreciate the difficulty in ensuring everyone is cited. However, there are a couple that should be added: (i) Regarding the dilation of the inner ring - as well as the references from Referee #2, Mahamid et al., 2016 should also be cited in this context as they were among the first to demonstrate this (even though a bit buried) and discuss: “Deviations from previous structures were evident for the pore diameter in the spoke ring (Fig. S7C)... This confirmed previous observations of NPC plasticity and illustrates the problem of analyzing large flexible structures by averaging methods”. Feldherr also published a couple of papers about the ability of NPCs to alter their functional diameter (e.g. Feldherr et al., 1998), which may also be of interest in this regard. (ii) Regarding potential membrane transport through gaps between the spokes – as well as the references from Referee #2, there is also Kim et al., 2018 that could also be cited in this context as they also saw these channels, stating: “This juxtaposition of arches and transient openings may delineate conduits for nucleocytoplasmic transport of transmembrane proteins, potentially resolving the issue of how membrane proteins transit the NPC”.

Response:

We thank the reviewer for their comments and have incorporated a reference to Mahamid et al. 2016 earlier in the text with regard to observations of large IR diameters as well as Feldherr et al. 1998 in the discussion about the impact of transport on IR diameter. Finally, as the reviewer suggested, we have included Kim et al. 2018 in our discussion of peripheral channels that may impact INM protein transport.

Referee #2 (Remarks to the Author):

The authors did an admirable job of addressing the major points raised in our review, improving their modeling and validation, as well as the discussion of previous in situ studies. The NPC model appears to be significantly improved and fits better into the experimental data, as shown in Extended Data Figure 3. Extended Data Figure 6 also now illustrates the manual classification of NPCs with missing nuclear and cytoplasmic rings. Although we believe that this study is nearly ready for publication, a few easily addressed, yet important, issues remain:

DISCUSSION OF PREVIOUS STUDIES:

Introduction, Lines 37-43: “NPCs from cryo-FIB milled single-cell organisms have been studied recently and show significant differences to human cells 28-30. HIV-1 infected human T cells also show differences 31. Here, we study the architecture of the human NPC from cryo-FIB milled DLD-1 cells. Our data establishes concepts that may have been previously interpreted as largely species-specific differences in the context of human cells. The organization of the IR increases the central pore diameter by more than 30%, while the two other ring moieties reorient differently with respect to the nuclear membrane.”

While we appreciate that the preceding studies are now mentioned in the introduction, two key pieces of information are missing that set the stage for the current work:

1) The preceding in situ NPC structures don't just show “differences”. It should be specifically mentioned in the introduction that all these studies observed a dilated inner ring diameter compared to structures from isolated NPCs. As far as the interpretations in the previous studies, both species-specific and physiological differences have been proposed for the dilated inner ring. The current study proves that it is the latter.

2) The study by Zila et al. (reference 31) observed equally dilated inner rings in both infected and non-infected human cells. Thus, the dilated inner ring has already been shown to be a feature of human cells, not a specific consequence of HIV infection. This is quantified in Fig. 6D of Zila et al. and described as follows: “To address if HIV-1 infection may promote NPC dilation, we collected cryo-electron tomograms of non-infected SupT1-R5 cells and measured the diameter of 39 NPCs. We found that the NPCs dilated to similar diameters in non-infected T cells (Figure 6D). Taken together, our findings indicate that the NPC structure observed under the relevant conditions, namely in infected and non-infected T cells in situ, is representative of the transporting state, whereas the constricted state observed in isolated nuclear envelopes (von Appen et al., 2015) may be more relevant to stress conditions (Zimmerli et al., 2020).”

While the authors are of course free to pick their own wording incorporating the above information, we suggest the following:

“NPCs from cryo-FIB milled yeast and algae cells have been studied recently and show significant differences to previous human NPC structures, including dilated IRs 28-30. Dilated IRs were also observed in cryo-FIB milled human T cells, both when infected with HIV and without infection 31. Here, we study the architecture of the human NPC from cryo-FIB milled DLD-1 cells. Our data

establishes concepts in human NPC architecture that may have previously been interpreted as species-specific differences. The dilated organization of the IR increases the central pore diameter by more than 30% compared to previous structures of isolated human NPCs, while the two other ring moieties reorient differently with respect to the nuclear membrane.”

Response:

We thank the reviewers for this feedback and have made the changes to our wording in the introduction. We reserved the phrase “dilated IR” in our manuscript to refer only to the change in IR dimensions upon Nup96-depletion, and instead use the word “wider IR” here.

The text now reads: NPCs from cryo-FIB milled yeast^{19,20}, algae²¹, and human cells^{22,23} show differences to previous human NPC structures, including wider IRs. Here, we study the architecture of the human NPC from cryo-FIB milled DLD-1 cells. Our data establishes concepts in human NPC architecture that may have previously been interpreted as species-specific differences. Our study shows that the cellular environment significantly influences the central diameter of the NPC and emphasizes the modular, though interdependent architecture of the NPC and its role in shaping the NE.

DATA AVAILABILITY:

The PDB file containing the atomic model coordinates generated by this study must be deposited at the PDB instead of Zenodo, following the Nature Editorial Policies as stated in:

<https://www.nature.com/nature/editorial-policies/reporting-standards#availability-of-data>
<https://www.nature.com/sdata/policies/repositories#molec>

“Mandatory deposition of data is required for certain data types; Macromolecular structure: Worldwide Protein Data Bank (wwPDB)”.

It is evident that a model generated from a map at this resolution level (~34 Å) should not be interpreted at atomic resolution. Nevertheless, the relative subunit positions and orientations are of major interest to the scientific community (in our original review, we called it “the current benchmark for native NPC architecture in human cells”). If the authors are concerned that their coordinates may be misinterpreted, they can for example trim the side chains from the model, leaving just the structure backbone, which is understandable at this resolution.

Response:

We deposited the models into the PDB as two entries to match the previous depositions. One that contains the CR and NR complex for a single protomer (PDB-7PEQ) and one that contains the IR complex (PDB-7PER). Both will be made available upon publication.

TYPOS:

Introduction, Line 40:

Our data establishes concepts that may have previously BEEN interpreted

Results, Line 172-174:

Although the resolution did not allow fitting the IR model into our map, we placed the model as an

estimate for the IR mass into these maps AND OBSERVED THAT the spacing between the IR subcomplexes increases as fewer ring structures are present (Fig. 4c).

Methods, visualization:
"chimeras" should be "Chimera's"

Response:

We have made the appropriate corrections.

A FINAL NOTE:

Finally, this does not affect our assessment of this paper, but we wanted to respond to the authors' claim that: "it is important to separate the discussion about human NPCs from those studies that involve other organisms. Three of the four studies mentioned involved *S. cerevisiae*, *S. pombe* and *C. reinhardtii*, respectively. These are single-cell organisms, each evolutionarily more than a billion years separated from humans."

While we appreciate the argument that there is great evolutionary distance between the different organisms that have been studied so far, there is nothing special about humans that make them above comparison to other organisms such as algae and yeast, especially for a molecular complex like the NPC, which is a shared defining feature of all eukaryotes. Multicellularity is the wrong metric—it is not indicative of evolutionary distance, and indeed it has arisen numerous times. In terms of genetic drift, algae and yeast are both more closely related to humans than they are to each other (please see the attached "phylogenies.pdf", which plots genetic distance across the tree of life). Unlike the more divergent outer rings of the NPC, the NPC inner ring is highly conserved between algae, yeast, and humans. Thus, the fact that both yeast and algae show a dilated inner ring in situ has accurately foreshadowed that the same would be true in humans. This conclusion is not intended to take anything away from the current study, but this is the clear evolutionary context that these new findings fit into.

Response:

We agree that multicellularity is perhaps not the best metric, but evolutionary distance is, which was our main argument. We also completely agree that no organism is 'above' or 'below' another organism, a nomenclature we have not used. All we wanted to point out is that we should be acutely aware of the actual data we have, and which kind of conclusions are likely and which ones are rather speculative. As nucleoporins are notoriously poorly conserved, and no one has yet observed subcomplex assemblies at sub-nanometer resolution, we believe that it is dangerous to make unifying statements based on data from very different organisms. Moreover, none of the species mentioned were structurally studied using purified nuclei, therefore changes in the NPC dimensions and membranes could be appreciated and attributed to the cellular environment. We meant to highlight that the comparisons our manuscript can make (i.e. human cells with human cells and mice cells with mice cells) remove any speculation about species-specificity or an impact of varying nucleoporin stoichiometry. We have done our best to explain this more precisely in the text.

Reviewed by Benjamin Engel and Ricardo Righetto

Referee #3 (Remarks to the Author):

The authors have addressed all points raised by the reviewers. The manuscript now clearly reports the state of current knowledge taking into account published work and describes adequately the knowledge gain of this study.

Response:

We thank the reviewer for their previous comments and are pleased we have addressed them completely.